# PEARL-PROX: PROXIMAL ALGORITHM FOR RESOLVING PLAYER DRIFT IN MULTIPLAYER FEDERATED LEARNING

## ABSTRACT

Recently, Yoon et al. (2025) introduced multiplayer federated learning (MpFL), a novel federated learning framework capable of formulating strategically behaving, rational clients. In MpFL, the clients are modeled as players of a multiplayer game with individual objectives, aiming to seek an equilibrium. While Per-Player Local Stochastic Gradient Descent (PEARL-SGD) algorithm has been proposed as a counterpart of Local SGD in the MpFL setup, it exhibits the *player drift* phenomenon—excessive local updates by individual players lead to divergence of the global dynamics. In this work, we formalize the concept of player drift and propose the *Per-Player Local Proximal Algorithm* (PEARL-Prox) to resolve it. PEARL-Prox lets each player optimize a regularized objective with high accuracy, ensuring convergence to the equilibrium while enabling the players to exploit their local compute budgets. Consequently, PEARL-Prox offers a significantly improved communication complexity of $\mathcal{O}\left(\log \epsilon^{-1}\right)$ compared to the $\Omega\left(\epsilon^{-1/2}\right)$ complexity of PEARL-SGD under the same theoretical assumptions.

## 1 INTRODUCTION

In classical federated learning (FL), multiple clients cooperatively train a model without revealing local data (Konečný et al., 2016a). It assumes that all clients' interests are aligned toward optimizing a shared global objective. However, strategic clients may have distinct objectives, potentially involving components that are competing with others' interests, which can be formulated in the language of game theory (Fudenberg & Tirole, 1991; Maschler et al., 2020). Recently, in Yoon et al. (2025), *multiplayer federated learning (MpFL)* has been proposed as a federated learning framework that formulates clients behaving as players in a game, with the collective goal of reaching the equilibrium.

One of the primary concerns in designing FL training algorithms is the frequency of *communication* (also referred to as synchronization or aggregation), the process by which the central server collects information from, and distributes updates to, each client (Konečný et al., 2016b). In FedAvg (Local SGD), the cornerstone of FL algorithms, each client locally performs SGD updates on their local data, which are aggregated by the server once in a while (McMahan et al., 2017). Due to its costly nature the communication occurs infrequently, which allows clients to perform a large number of local computations between communications. However, letting each client fully exploit their computational capabilities to minimize (run a local optimization algorithm for too many steps) their local objectives may lead to suboptimal results (McMahan et al., 2017; Mitra et al., 2021). In classical FL, this phenomenon is called *client drift*, and occurs when the global model fails to converge to the minimizer of the global objective due to data heterogeneity (Zhao et al., 2018; Li et al., 2020a). In recent years, multiple works have proposed algorithmic adjustments to mitigate client drift (Li et al., 2020a; Karimireddy et al., 2020; Gorbunov et al., 2021; Mishchenko et al., 2022).

The analogous phenomenon to the client drift, in the setup of MpFL, is called the *player drift* (Yoon et al., 2025)—greedy local optimization by each player causes the game dynamics to diverge away from the equilibrium. However, as we explain in this work the player drift arises not only from heterogeneity in data (like classical FL) but also due to the conceptually different setting of the game dynamics appear in MpFL—it is attributed to the nature of multiplayer games where the players have possibly competing objectives. The first MpFL algorithm Per-Player Local SGD (PEARL-SGD), proposed in Yoon et al. (2025), is prone to the player drift phenomenon.

In this work, we propose PEARL-Prox, an algorithm for MpFL where each player locally optimizes their objective, but with a regularizer. This allows each player to leverage their local computing power to reach the optimum, while ensuring the global game dynamics to converge to the equilibrium.

## 1.1 MAIN CONTRIBUTIONS

⋄ **Formalizing player drift.** We mathematically formalize the notion of *player drift*, the phenomenon where an MpFL algorithm diverges if each player takes an excessive number of local steps. We capture that the major cause of player drift is not simply data heterogeneity but the game-theoretic structure of the underlying problem where greedy minimization of the local objective by each player leads to collectively inferior dynamics.

⋄ PEARL-Prox **algorithm for mitigating player drift.** We propose *Per-Player Local Proximal Algorithm* (PEARL-Prox), a new algorithm for MpFL where each player minimizes a quadratically regularized objective in each local update round between communication steps. PEARL-Prox converges when all players reach the minimum of the regularized objectives unlike PEARL-SGD, which resolves player drift and allows players to fully exploit the local computation budgets. In the stochastic setup, and if large number of local iterations are allowed, the communication cost can be reduced by a much larger factor compared to PEARL-SGD.

⋄ **Convergence guarantees for** PEARL-Prox**.** We analyze the convergence of PEARL-Prox for distinct setups, providing the following results:

- In the idealized scenario where proximal operations (minima computation for regularized objectives) can be performed exactly, PEARL-Prox converges linearly to the equilibrium.
- If proximal operations are performed with errors, PEARL-Prox li nearly converges to the neighborhood of the equilibrium whose radius is proportional to the sum of the errors. This holds regardless of the subroutine (algorithm) used by each player for approximate proximal computations.
- We provide a concrete convergence result and analyze the communication complexity for a version of PEARL-Prox using SGD as a subroutine.

⋄ **Numerical experiments.** We verify our theoretical results via numerical experiments, demonstrating the effectiveness and practicality of the proposed PEARL-Prox in handling player drift.

## 2 MPFL: FORMULATION AND RELATED FL TOPICS

MpFL is a recently introduced federated learning framework capable of handling a broad scope of setups expressible using game-theoretic formulation. While its development is currently focused on the theory, in this section we illustrate the intersection of MpFL with several seemingly distinct, popular FL frameworks, highlighting its wide applicability. In Appendix A, we further discuss related literature with more detailed explanation of heterogeneity and client/player drift.

### 2.1 FORMULATION AND NOTATIONS FOR MPFL

We consider the multiplayer game setup where there are $n$ players, indexed by $i = 1, \ldots, n$. The player $i$ has an unconstrained and continuous space of actions $x_i \in \mathbb{R}^{d_i}$. We denote the joint action vector of all players by $\boldsymbol{x} = (x_1, \ldots, x_n) \in \mathbb{R}^D = \mathbb{R}^{d_1 + \cdots + d_n}$. Each player $i$ has their objective function $f_i(x_1, \ldots, x_n) \colon \mathbb{R}^{d_1 + \cdots + d_n} \to \mathbb{R}$ which they prefer to minimize with respect to $x_i$. The goal of an $n$ player (multiplayer) game is to find an *equilibrium* $\boldsymbol{x}^\star = (x_1^\star, \ldots, x_n^\star) \in \mathbb{R}^D$, which formally expressed as

$$\underset{\boldsymbol{x}^\star = (x_1^\star, \ldots, x_n^\star) \in \mathbb{R}^D}{\text{find}} \quad f_i(x_i^\star; x_{-i}^\star) \leq f_i(x_i; x_{-i}^\star), \quad \forall x_i \in \mathbb{R}^{d_i}, \quad \forall i \in [n], \tag{1}$$

where $x_{-i} = (x_1, \ldots, x_{i-1}, x_{i+1}, \ldots, x_n) \in \mathbb{R}^{D-d_i}$ denotes the vector of actions from all players other than the player $i$, and $f_i(x_i; x_{-i}) = f_i(x_1, \ldots, x_n)$.

MpFL considers the setup where $f_i$ are given as $f_i(x_1, \ldots, x_n) = \mathbb{E}_{\xi_i \sim \mathcal{D}_i}[f_{i,\xi_i}(x_1, \ldots, x_n)]$ where $\mathcal{D}_i$ is the data distribution for the player $i$, $\xi_i$ are data samples and $f_{i,\xi_i}(\cdot)$ is the corresponding loss function. Given fixed $x_{-i}$, each player can compute the stochastic gradients $\nabla_{x_i} f_{i,\xi_i}(\cdot; x_{-i})$ of $f_i$. Based on these computations, in each round (before communication occurs), each player locally adjusts their action by the amount $\Delta x_i$. Then at the communication step, the server collects these updated local actions $(x_1 + \Delta x_1, \ldots, x_n + \Delta x_n)$ and distributes them back to all players.

## 2.2 CLOSELY RELATED FL FRAMEWORKS: SIMILARITIES AND DIFFERENCES

Below, we discuss connections between MpFL and personalized FL, vertical FL and federated/multiagent reinforcement learning.

**Personalized federated learning.** In personalized federated learning (PFL), clients cooperate as in FL, but with the goal of training local models $x^i$, performing well on their local dataset $\mathcal{D}_i$ (rather than producing a single global model) (Hanzely & Richtárik, 2020; T. Dinh et al., 2020). Multiple approaches such as mixing the local and global models (Hanzely & Richtárik, 2020; Deng et al., 2020), model-agnostic meta-learning (MAML) based formulation (Fallah et al., 2020) or bilevel optimization formulation using Moreau envelopes (T. Dinh et al., 2020) have been proposed and gained popularity. Among these, consider the following characterization of PFL which does not require to explicitly keep a separate global model apart from local models (Hanzely & Richtárik, 2020; Hanzely et al., 2020; 2023):

$$\operatorname*{minimize}_{\boldsymbol{x}=(x^1,\ldots,x^n)\in\mathbb{R}^{nd}} \quad \frac{1}{n}\sum_{i=1}^{n} h_i(x_i) + \frac{\alpha}{2n}\sum_{i=1}^{n}\|x_i - \overline{x}\|^2,$$

where $x_1, \ldots, x_n \in \mathbb{R}^d$ are local models of each player, $h_1, \ldots, h_n$ are the local losses following each player's distributions, $\overline{x} = \frac{1}{n}\sum_{i=1}^{n} x_i$ is the average model and $\alpha > 0$ is a hyperparameter. As the first-order optimality condition to the above problem is given by $x_i^\star - \overline{x}^\star + \frac{1}{\alpha}\nabla h_i(x_i^\star) = 0$ (Hanzely et al., 2020), when each $h_i$ is convex, this is equivalent to finding an equilibrium of the $n$-player game where each player has the objective function $f_i(x_i; x_{-i}) = h_i(x_i) + \frac{\alpha}{2}\sum_{i=1}^{n}\|x_i - \overline{x}\|^2$. That is, one of the popular formulations for PFL can be viewed as an instance of MpFL problem.

**Vertical federated learning.** Classical FL implicitly assumes the "horizontal" FL setting, where each client holds data of distinct users but with identical features (Yang et al., 2019). In the vertical federated learning (VFL) setting, multiple clients rather possess data of distinct features for the identical set of users; in this scenario, the clients require a coordinated collaboration both in the training phase and in the inference phase (Liu et al., 2024). We formally describe the formulated provided in Liu et al. (2022): the $n$ clients have the dataset with $N$ data $\mathcal{D} = \{\phi_j, y_j\}_{j=1}^N$, consisting of feature vectors $\phi_j = (\phi_{j,1}, \ldots, \phi_{j,n}) \in \mathbb{R}^{m_1 + \cdots + m_n}$ and the corresponding classification label $y_j$. Client $i$ has access to the $i$-th 'block feature' $\phi_{j,i} \in \mathbb{R}^{m_i}$ for each data $j = 1, \ldots, N$, i.e., has the dataset $\mathcal{D}_i = \{\phi_{j,i}\}_{j=1}^N$. Each client has a model parameter $x_i \in \mathbb{R}^{d_i}$, and they collaborate to solve the problem

$$\operatorname*{minimize}_{(x_1,\ldots,x_n)\in\mathbb{R}^D} \frac{1}{N}\sum_{j=1}^{N} \ell(x_1,\ldots,x_n;\phi_j,y_j) + \alpha\sum_{i=1}^{n} g(x_i)$$

where $\ell$ is a loss function (depending on model parameters of all clients), $g\colon \mathbb{R}^{d_i} \to \mathbb{R}$ are client-wise regularizers, $\alpha > 0$ is a hyperparameter and $D = d_1 + \cdots + d_n$. This scenario can be viewed as a multiplayer game where each player $i \in [n]$ has the objective function $f_i(x_i; x_{-i}) = \frac{1}{N}\sum_{j=1}^{N} \ell(x_1,\ldots,x_n;\phi_j,y_j) + \alpha g(x_i)$. One can verify that PEARL-SGD applied to this setup (viewing the setting as MpFL) gives the update rule identical to that of the FedBCD algorithm in Liu et al. (2022). However, we note that VFL applications generally require additional systematic components, e.g. homomorphic encryption (Rivest et al., 1978) to keep $x_i$ private, while in MpFL, communication efficiency is of primary interest and privacy is considered auxiliary, or orthogonal.

**Multiagent reinforcement learning.** In multiagent reinforcement learning (MARL), $n$ agents interact within a common environment with the set of states $\mathcal{S}$. Each agent $i$ has its action space $\mathcal{A}_i$, and tries to optimize a policy $\pi_i\colon \mathcal{S} \to \Delta(\mathcal{A}_i)$. Given a state $s$ and the joint actions $\boldsymbol{a} = (a_1, \ldots, a_n)$ of all players, the environment yields a next state $s_{t+1}$ according to a distribution $P(\cdot|s_t, a_t)$, and rewards $R_i(s_t, \boldsymbol{a}_t, s_{t+1})$ for each agent. The value function of the agent $i$ is defined as

$$V_{\pi_i,\pi_{-i}}(s) = \mathbb{E}_{s_{t+1}\sim P(\cdot|s_t,a_t), a_{-i}\sim\pi_{-i}(\cdot|s_t)}\left[\sum_{t\geq 0}\gamma^t R_i(s_t, \boldsymbol{a}_t, s_{t+1})\,\middle|\, a_{i,t}\sim\pi_i(\cdot|s_t), s_0 = s\right]$$

---

**Algorithm 1** PEARL-SGD

---

**Input:** Step sizes $\{\gamma_k^p\}_{k=0}^{\tau-1} > 0$, Synchronization interval $\tau \geq 1$, Number of synchronization/local update rounds $R \geq 1$

    **for** $p = 0, \ldots, R-1$ **do**
        Master server collects $x_i^p$ from players $i = 1, \ldots, n$ and forms $\boldsymbol{x}^p = (x_1^p, \ldots, x_n^p)$
        Master server distributes $\boldsymbol{x}^p$ back to players $i = 1, \ldots, n$
        **for** $i = 1, \ldots, n$ **do**
            $x_i^{p+1} \leftarrow \text{SGD}\left(f_i(\cdot\,; x_{-i}^p), x_i^p, \tau, \{\gamma_k^p\}_{k=0}^{\tau-1}\right)$
        **end for**
    **end for**

**Output:** $\boldsymbol{x}^R \in \mathbb{R}^D$

---

The concept of Nash equilibrium among agents is characterized by $V_{\pi_i^\star, \pi_{-i}^\star}(s) \geq V_{\pi_i, \pi_{-i}^\star}(s), \forall s \in \mathcal{S}$ (Zhang et al., 2021) for any policy $\pi_i$, $i = 1, \ldots, n$. The significant challenge of solving this problem arises from both its game-theoretic structure and non-stationarity of the environment from the perspective of each agent (Lowe et al., 2017; Al-Shedivat et al., 2018; Qi et al., 2021). One interpretation of the problem is that it seeks Nash equilibria for $|\mathcal{S}|$ distinct games, one for each state $s \in \mathcal{S}$. MpFL does not immediately subsume this formulation in its basic form, but there exists a close connection which could be made precise by extending the framework of MpFL using multiobjective games (Zhao, 1991), where $\pi_i$ being the decision variable (action) of the player $i$ and objective functions are value functions with one component per each $s \in \mathcal{S}$.

## 3 PLAYER DRIFT IN MPFL

In this section, we define and discuss player drift. Let us first consider the PEARL-SGD (Algorithm 1) from Yoon et al. (2025), designed for the MpFL setup. In PEARL-SGD, at each communication round $p = 0, \ldots, R-1$, each player $i$ is given with $\boldsymbol{x}^p = (x_1^p, \ldots, x_n^p)$. Then they run an inner loop of SGD with $f_i(\cdot\,; x_{-i}^p)$ as the objective function, $x_i^p$ as the initial point, and perform $\tau$ steps of stochastic gradient descent using a step size sequence $\{\gamma_k^p\}_{k=0}^{\tau-1}$. The output of this SGD subroutine $x_i^{p+1}$ is then sent to the server for synchronization.

---

**Algorithm 2** $\text{SGD}(h, x^0, T, \{\gamma_k\}_{k=0}^{T-1})$

---

**Input:** Objective function $h(x) = \mathbb{E}_{\xi \sim \mathcal{D}}[h_\xi(x)]$, initial point $x^0$, total iteration number $T$, step sizes $\{\gamma_k\}_{k=0}^{T-1}$

    **for** $k = 0, \ldots, T-1$ **do**
        Sample $\xi_k \sim \mathcal{D}$.
        $x^{k+1} \leftarrow x^k - \gamma_k \nabla h_{\xi_k}(x^k)$
    **end for**

---

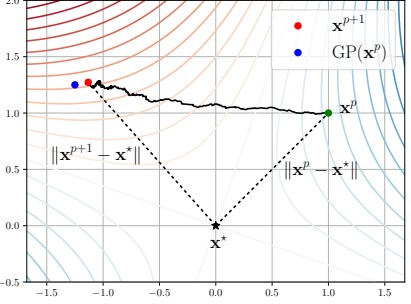

Figure 1: PEARL-SGD behaving similarly to GP dynamics for minimax game with $f(x_1, x_2) = \frac{\mu}{2}x_1^2 + x_1 x_2 - \frac{\mu}{2}x_2^2, \mu \in (0, 1)$.

**Greedy player dynamics.** Suppose that the MpFL environment has low communication frequency and players have sufficient computation budget for local optimization (i.e., $\tau$ is large enough). If each player acts selfishly, they can select an appropriate sequence of step sizes to minimize their local cost function $f_i$. In other words, $\lim_{\tau \to \infty} x_i^{p+1}(\tau; \boldsymbol{x}^p) = \arg\min_{x_i \in \mathbb{R}^{d_i}} f_i(\cdot\,; x_{-i}^p) := \tilde{x}_i^\star(x_{-i}^p)$, where we omit the dependency on $\gamma_k^p$'s, assuming they can be optimally tuned. Then the resulting global dynamics between adjacent synchronization steps can be expressed as

$$\boldsymbol{x} \mapsto \text{GP}(\boldsymbol{x}) := (\tilde{x}_1^\star(x_{-1}), \ldots, \tilde{x}_n^\star(x_{-n}))$$

which we call the *greedy player (GP)* dynamics.

However, even under favorable assumptions such as strong monotonicity of $F$, the greedy player dynamics may diverge away from the equilibrium, i.e., one can have $\|\text{GP}(\boldsymbol{x}) - \boldsymbol{x}^\star\|^2 / \|\boldsymbol{x} - \boldsymbol{x}^\star\|^2 > 1$ (Figure 1). We say that *player drift* occurs if excessive local compute in an MpFL algorithm results in a similar divergent behavior of the GP dynamics, i.e., $\lim_{\tau \to \infty} \|\boldsymbol{x}^{p+1}(\tau; \boldsymbol{x}^p) - \boldsymbol{x}^\star\|^2 / \|\boldsymbol{x}^p - \boldsymbol{x}^\star\|^2 > 1$.

**Algorithm 3** Per-Player Local Proximal Algorithm (PEARL-Prox)

---

**Input:** Regularization parameter $\lambda > 0$, Number of rounds $R \geq 1$

    Initialize $x^0 = (x_1^0, x_2^0, \ldots, x_n^0)$

    **for** $p = 0, \ldots, R - 1$ **do**

        Master server collects $x_i^p$ from players $i = 1, \ldots, n$ and forms $\boldsymbol{x}^p = (x_1^p, \ldots, x_n^p)$

        Master server distributes $\boldsymbol{x}^p$ back to players $i = 1, \ldots, n$

        **for** $i = 1, \ldots, n$ **do**

            Approximately compute $x_i^{p+1} \approx \arg\min_{x_i} f_i(x_i; x_{-i}^p) + \frac{\lambda}{2} \|x_i - x_i^p\|^2$

        **end for**

    **end for**

**Output:** $x^R \in \mathbb{R}^D$

---

In particular, in Yoon et al. (2025), it was shown that PEARL-SGD converges to $\boldsymbol{x}^\star$, but only if the step size $\gamma = \mathcal{O}(1/\tau)$ is used. This indicates that the net progression allowed per local computation round between synchronization steps should be limited. In this regard, PEARL-SGD suffers from player drift, as the local compute power of players (clients) cannot be fully exploited.

## 4 PEARL-Prox: ALGORITHM DESIGN AND ANALYSIS

In this section, we propose the new algorithm PEARL-Prox for handling the player drift in MpFL. In each communication round of PEARL-Prox, players minimize the regularized local objectives $f_i(\cdot; x_{-i}^p) + \frac{\lambda}{2} \|\cdot - x_i^p\|^2$ with hyperparameter $\lambda > 0$, instead of $f_i(\cdot; x_{-i}^p)$.

When each player has sufficient compute power, at each round, they will attain the solution

$$
\begin{aligned}
x_i^{p+1} &= \mathrm{Prox}_{\frac{1}{\lambda} f_i(\cdot; x_{-i}^p)}(x_i^p) \\
&= \arg\min_{x_i \in \mathbb{R}^{d_i}} f_i(\cdot; x_{-i}^p) + \frac{\lambda}{2} \|x_i - x_i^p\|^2.
\end{aligned}
\tag{2}
$$

With $\lambda$ chosen appropriately, this identifies an update direction which points toward the equilibrium $\boldsymbol{x}^\star$ but is not excessive (as the objective enforces $x_i^{p+1}$ to be close to $x_i^p$), and thus, convergence is ensured. Importantly, one

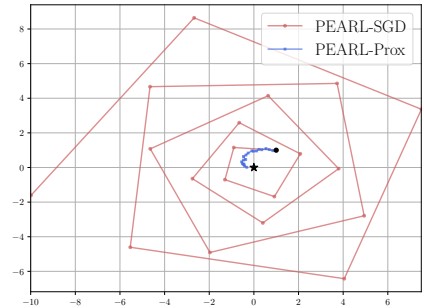

Figure 2: Trajectory plot for the minimax game with $f(x_1, x_2) = \frac{\mu}{2}x_1^2 + x_1 x_2 - \frac{\mu}{2}x_2^2$, $\mu = 0.8$. While PEARL-SGD diverges with $\gamma_k^p \equiv \gamma = 0.1$, $\tau = 25$, PEARL-Prox converges (with $\lambda = 10$).

does not have to specify a subroutine used by each player to minimize the regularized objectives, including the choice of algorithm, number of iterations or algorithm parameters such as step sizes. Algorithm 3 emphasizes this aspect with the intentional ambiguity.

As a result of the algorithm design, PEARL-Prox can take the full advantage of the local steps and resolve player drift. Unlike PEARL-SGD where minimization of $f_i(\cdot; x_{-i}^p)$ can cause divergence, in PEARL-Prox accurate minimization is rather beneficial and enables convergence (Figure 2, Theorem 4.4). Due to this characteristic, in the stochastic setting, PEARL-Prox provides a much larger communication gain compared to the case of PEARL-SGD (see Section 4.4).

### 4.1 THEORETICAL ASSUMPTIONS

We present the assumptions required for convergence analyses. First we assume local objectives are conducive to minimization (smooth and convex), following the standards of first order optimization.

**Assumption 4.1.** For $i = 1, \ldots, n$, for any $x_{-i} \in \mathbb{R}^{D-d_i}$, the function $f_i(\cdot; x_{-i}): \mathbb{R}^{d_i} \to \mathbb{R}$ is convex and $L_i$-smooth, i.e., $\nabla_{x_i} f_i(\cdot; x_{-i})$ is $L_i$-Lipschitz continuous.

We define $F: \mathbb{R}^D \to \mathbb{R}^D$ by $F(\boldsymbol{x}) = F(x_1, \ldots, x_n) = (\nabla_{x_1} f_1(x_1; x_{-1}), \ldots, \nabla_{x_n} f_n(x_n; x_{-n}))$, which we call the *joint gradient operator*. Given Assumption 4.1, the equilibrium condition (1) is equivalent to $F(\boldsymbol{x}^\star) = 0$. For the simultaneous gradient descent (gradient play) algorithm to converge, the literature on learning in games or distributed Nash equilibrium search commonly assumes the existence of an equilibrium that is (globally) variationally stable (Mertikopoulos & Zhou,

2019; Mertikopoulos et al., 2024) or the similar (yet stronger) restricted/quasi-strong monotonicity (Loizou et al., 2021; Tatarenko et al., 2021; Yoon et al., 2025). Following these lines of works, to establish quantitative convergence guarantees, we assume the quasi-strong monotonicity.

While Assumption 4.1 implies that $f_i$ is smooth in the $x_i$-coordinate, it does not imply any smoothness property with respect to $x_{-i}$, which is necessary for proving convergence to equilibrium in games. Many prior works assume Lipschitz continuity of $F$, which, together with quasi-strong monotonicity, implies the cocoercivity respect to $x^\star$ (called star-cocoercivity). Here, following the presentations of Loizou et al. (2021); Yoon et al. (2025), we directly assume this weaker property instead.

**Assumption 4.2.** The joint gradient operator $F$ is $\mu$-quasi-strongly monotone, i.e., there exists an equilibrium $x^\star \in \mathbb{R}^D$ and $\mu > 0$ such that $\forall x \in \mathbb{R}^D$, $\langle F(x), x - x^\star \rangle \geq \mu \|x - x^\star\|^2$. Further, $F$ is $\ell$ star-cocoercive, i.e., there exists $\ell > 0$ such that $\forall x \in \mathbb{R}^D$, $\langle F(x), x - x^\star \rangle \geq \ell \|F(x)\|^2$.

Finally, as optimization of the local objectives $f_i$ will generally be stochastic, we include the following bounded variance assumption on the stochastic gradients.

**Assumption 4.3.** For $i = 1, \ldots, n$, there exists $\sigma_i \geq 0$ such that for any $x_i \in \mathbb{R}^{d_i}, x_{-i} \in \mathbb{R}^{D-d_i}$, we have $\mathbb{E}_{\xi_i \sim \mathcal{D}_i}\left[\|\nabla_{x_i} f_{i,\xi_i}(x_i; x_{-i}) - \nabla_{x_i} f_i(x_i; x_{-i})\|^2\right] \leq \sigma_i^2$.

Note that Assumptions 4.1 through 4.3 is the same set of assumptions as in Yoon et al. (2025).

### 4.2 Convergence of exact PEARL-Prox

To illustrate the game dynamics expected as a result of minimizing the regularized objectives, we first analyze the idealized scenario where we assume that each player is capable of computing (2) exactly. In this case, PEARL-Prox converges linearly to $x^\star$, as deterministic (full-batch) version of PEARL-SGD does (Yoon et al., 2025, Theorem 3.3).

**Theorem 4.4.** Suppose that Assumptions 4.1 and 4.2 hold. Let $L_{\max} = \max\{L_1, \ldots, L_n\}$, and let $\kappa = \ell/\mu$ be the condition number of the game. Suppose $\lambda > \frac{1}{2}(\ell + 2L_{\max}\sqrt{\kappa})$. Then PEARL-Prox with exact proximal operator computation (2) converges with the rate

$$\|x^R - x^\star\|^2 \leq \left(1 - \frac{2\mu\zeta}{\lambda}\right)^R \|x^0 - x^\star\|^2$$

where $\zeta = 1 - \frac{\ell + 2L_{\max}\sqrt{\kappa}}{2\lambda} > 0$.

Theorem 4.4 requires $\lambda$ to be sufficiently large, which is intuitively clear as exact PEARL-Prox will converge to the GP dynamics when $\lambda \to 0$. In other words, if $\lambda$ is small, the effect of algorithmic modification of adding the quadratic regularizer will not be strong enough to fix player drift.

### 4.3 Convergence of PEARL-Prox with inexactness

We have observed that PEARL-Prox converges with exact proximal operator computation, if $\lambda$ is chosen appropriately. In practice, however, players have finite compute limits and will be only able to compute (2) with some error (inexactness). Below, we provide a convergence guarantee for this scenario. For that purpose, we start with quantifying the magnitude of errors.

**Assumption 4.5.** (Magnitude of errors) At each round $p = 0, \ldots, R - 1$, each player $i = 1, \ldots, n$ computes $x_i^{p+1} = \text{Prox}_{\frac{1}{\lambda} f_i(\cdot; x_{-i}^p)}(x_i^p) + v_i^p$ for some random error vector $v_i^p \in \mathbb{R}^{d_i}$ satisfying

$$\mathbb{E}\left[\|v_i^p\|^2 \Big| x_i^p\right] \leq \delta_i^p \|\nabla_{x_i} f_i(x_i^p; x_{-i}^p)\|^2 + \epsilon_i^p \tag{3}$$

for some $\delta_i^p, \epsilon_i^p > 0$, where the conditional expectation is taken over all randomness within the process of computing $x_i^{p+1}$ given $x_i^p$.

We consider the error bound of the form (3) because most optimization algorithms minimizing a strongly convex function $h(z)$ exhibit a convergence guarantee of the form

$$\mathbb{E}\left[\|z^T - z^\star\|^2\right] \leq a_T \|z^0 - z^\star\|^2 + b_T \tag{4}$$

where $T$ is the total iteration number, $z^0$ is the initial point and $z^\star = \arg\min_z h(z)$. In our context, each player $i$, at round $p$, takes $f_i(\cdot; x_{-i}^p) + \frac{\lambda}{2} \|\cdot - x_i^p\|^2$ as the objective function (whose minimum is $\text{Prox}_{\frac{1}{\lambda} f_i(\cdot; x_{-i}^p)}(x_i^p)$), $x_i^p$ as the initial point, and execute a minimization subroutine of their choice to output the approximate minimizer $x_i^{p+1}$. In this setting, the left hand side of (4) will be $\mathbb{E}\left[\|v_i^p\|^2 \big| x_i^p\right]$.

The $\|z^0 - z^\star\|^2 = \left\|x_i^p - \text{Prox}_{\frac{1}{\lambda} f_i(\cdot; x_{-i}^p)}(x_i^p)\right\|^2$ term in the right hand side can be upper bounded using the following Lemma 4.6. Combined, (4) can be rewritten into the form (3).

**Lemma 4.6.** Suppose $f_i(\cdot; x_{-i}^p) \colon \mathbb{R}^{d_i} \to \mathbb{R}$ is convex and let $\lambda > 0$. Then

$$\left\|x_i^p - \text{Prox}_{\frac{1}{\lambda} f_i(\cdot; x_{-i}^p)}(x_i^p)\right\| \leq \frac{1}{\lambda} \left\|\nabla_{x_i} f_i(x_i^p; x_{-i}^p)\right\|.$$

With the above components, we are now ready to state the convergence result.

**Theorem 4.7.** Suppose that Assumptions 4.1 and 4.2 hold. Suppose that $\lambda > 2\left(\ell + L_{\max}\sqrt{\kappa}\right)$ ($L_{\max}, \kappa$ defined as in Theorem 4.4) and PEARL-Prox satisfies Assumption 4.5 with $\delta_i^p \leq \frac{1}{4\lambda\ell}$ for $i = 1, \ldots, n$ and $p = 0, \ldots, R-1$. Then PEARL-Prox exhibits the rate

$$\mathbb{E}\left[\|\boldsymbol{x}^R - \boldsymbol{x}^\star\|^2\right] \leq \left(1 - \frac{\mu\zeta}{\lambda}\right)^R \|\boldsymbol{x}^0 - \boldsymbol{x}^\star\|^2 + \left(2 + \frac{2\lambda}{\mu}\right) \sum_{p=0}^{R-1} \left(1 - \frac{\mu\zeta}{\lambda}\right)^{R-p-1} \sum_{i=1}^{n} \epsilon_i^p$$

where $\zeta = 1 - \frac{2\left(\ell + L_{\max}\sqrt{\kappa}\right)}{\lambda} > 0$.

Theorem 4.7 implies that if each player minimizes $f_i(\cdot; x_{-i}^p) + \frac{\lambda}{2} \|\cdot - x_i^p\|^2$ with a certain degree of accuracy, then PEARL-Prox linearly converges to a correspondingly sized neighborhood of the equilibrium. The technical condition $\delta_i^p \leq \frac{1}{4\lambda\ell}$ can be fulfilled easily if each player uses a moderately large local iteration number, as $\delta_i^p$ corresponds to $a_T$ in the general guarantee (4) which decays exponentially in $T$ for strongly convex objectives.

**Remark.** Theorem 4.7 generalizes Theorem 4.4 in the sense that the case $\delta_i^p = \epsilon_i^p = 0$ (indicating exact prox computation) reduces to a linear convergence result as in Theorem 4.4. However, the condition on $\lambda$ and the linear convergence factor become more conservative due to some technical steps introduced to handle stochasticity.

## 4.4 PEARL-Prox WITH SGD INNER LOOP: CONVERGENCE & COMMUNICATION EFFICIENCY

In principle, players can use any algorithm to compute the approximate minimizer of the function $f_i(\cdot; x_{-i}^p) + \frac{\lambda}{2} \|\cdot - x_i^p\|^2$. Nevertheless, in this section, we provide a concrete, implementable version of PEARL-Prox (Algorithm 4) where all players use SGD as their minimization subroutine. This will convey a general idea of how players can select their algorithm parameters to converge to the equilibrium, and allow us to make a direct comparison to PEARL-SGD in terms of communication complexity. Let us denote the number of SGD steps taken by each player $i$ at round $p$ by $\tau_i^p$, and the corresponding step-size schedule by $\gamma_{i,t}^p$ ($t = 0, \ldots, \tau_i^p - 1$). Then we have:

---

**Algorithm 4 PEARL-Prox** with SGD subroutine

**Input:** Regularization parameter $\lambda > 0$, Number of rounds $R \geq 1$, SGD parameters $\tau_i^p, \{\gamma_{i,t}^p\}_{t=0}^{\tau_i^p}$

   Initialize $x^0 = (x_1^0, x_2^0, \ldots, x_n^0)$
   **for** $p = 0, \ldots, R-1$ **do**
      Master server collects $x_i^p$ from players $i \in [n]$
      Master server distributes $\boldsymbol{x}^p$ back to players
      **for** $i = 1, \ldots, n$ **do**
         $x_i^{p+1} \leftarrow \text{SGD}\big(f_i(\cdot; x_{-i}^p) + \frac{\lambda}{2} \|\cdot - x_i^p\|^2, x_i^p,$
         $\tau_i^p, \{\gamma_{i,t}^p\}_{t=0}^{\tau_i^p}\big)$
      **end for**
   **end for**
**Output:** $x^R \in \mathbb{R}^D$

---

**Corollary 4.8.** Suppose that Assumptions 4.1, 4.2 and 4.3 hold and $\lambda > 2\left(\ell + L_{\max}\sqrt{\kappa}\right)$. Let $\zeta = 1 - \frac{2\left(\ell + L_{\max}\sqrt{\kappa}\right)}{\lambda} > 0$, $\tau \geq \max\left\{\sqrt{\frac{4\ell}{\lambda}}, 16\left(1 + \frac{L_{\max}}{\lambda}\right)^2\right\}$ and $\sigma^2 = \sum_{i=1}^{n} \sigma_i^2$. Then

Algorithm 4 with $\tau_i^p \equiv \tau$ and $\gamma_{i,t}^p \equiv \frac{2\log\tau}{\lambda\tau}$ converges with the rate

$$\mathbb{E}\left[\left\|\boldsymbol{x}^R - \boldsymbol{x}^\star\right\|^2\right] \leq \left(1 - \frac{\mu\zeta}{\lambda}\right)^R \left\|\boldsymbol{x}^0 - \boldsymbol{x}^\star\right\|^2 + \left(2 + \frac{2\lambda}{\mu}\right)\frac{2\sigma^2\log\tau}{\mu\zeta\lambda\tau} \tag{5}$$

Note that as we use SGD as a subroutine for computing the proximal operator, we are interested in moderately large $\tau$, sufficient for attaining certain level of accuracy. This is why we have a lower bound on $\tau$ in Corollary 4.8. Nevertheless, due to the condition $\lambda > 2(\ell + L_{\max}\sqrt{\kappa})$ this lower bound is not too restrictive; e.g., with $\lambda = 4(\ell + L_{\max}\sqrt{\kappa})$ we have $\max\left\{\sqrt{\frac{4\ell}{\lambda}}, 16\left(1 + \frac{L_{\max}}{\lambda}\right)^2\right\} \leq 25$.

**Communication efficiency.** Note that (5) indicates linear convergence to a neighborhood with size $\propto \frac{\log\tau}{\tau}$, which is reduced as $\tau$ grows larger. Importantly, this does not depend on the number of communication $R$ and clearly indicates the advantage of large $\tau$. In particular, if $\tau$ is sufficiently large ($\tau = \tilde{\Omega}\left(\epsilon^{-1}\right)$), we have $\mathbb{E}\left[\left\|\boldsymbol{x}^R - \boldsymbol{x}^\star\right\|^2\right] \leq \epsilon$ if $R \gtrsim \frac{\lambda}{\mu\zeta}\log\frac{\left\|\boldsymbol{x}^0 - \boldsymbol{x}^\star\right\|^2}{\epsilon} = \Omega\left(\frac{\ell + L_{\max}\sqrt{\kappa}}{\mu}\log\frac{1}{\epsilon}\right)$. That is, even in the stochastic setup, PEARL-Prox can take full advantage of large local computation budget of the players to compute proximal operator with high accuracy, and reach the $\epsilon$-neighborhood of the equilibrium using only logarithmically many communications, which is comparable to the complexity from deterministic (full-batch) scenario. This is in sharp contrast with the case of PEARL-SGD, which requires at least $\Omega\left(\epsilon^{-1/2}\right)$ communication rounds in the stochastic setup, regardless of how large $\tau$ is. For a more detailed argument on this, we refer the readers to Appendix C.

## 5 NUMERICAL EXPERIMENTS

In this section, we present numerical experiments, with the focus of validating our theoretical predictions. We additionally illustrate the advantages of PEARL-Prox in terms of the player drift.

Following the setup in Yoon et al. (2025), we consider $n$-player game with $d_1 = \cdots = d_n = d$ and

$$f_i(x_i; x_{-i}) = \frac{1}{M}\sum_{m=1}^{M}\frac{1}{2}x_i^\mathsf{T} A_{i,m}x_i + \sum_{\substack{1\leq j\leq n \\ j\neq i}} x_i^\mathsf{T} B_{i,j,m}x_j + c_{i,m}^\mathsf{T} x_i \tag{6}$$

where $A_{i,m}, B_{i,j,m}$ are $d \times d$ symmetric matrices and $c_{i,m} \in \mathbb{R}^d$ for $i, j = 1, \ldots, n, j \neq i$, generated randomly for $m = 1, \ldots, M$. Here the randomness $\xi_i \sim \mathcal{D}_i$ is implemented by mini-batching from the finite sum. To ensure that (6) fulfills our theoretical assumptions, we generate random $A_{i,m}, B_{i,j,m}$ whose spectra are respectively within $[\mu_A, L_A]$ $(0 < \mu_A \leq L_A)$ and $[0, L_B]$ $(L_B > 0)$, and take $B_{j,i,m} = -B_{i,j,m}^\mathsf{T}$ for $i \neq j$ and $m = 1, \ldots, M$. We use $n = 5, d = 10, \mu_A = 10^{-2}, L_A = 1$ and use minibatches of size 10 for stochastic gradients. We use $M = 1000, L_B = 5$ for the first experiment and $M = 100, L_B = 10$ for the rest.

**Communication efficiency of PEARL-Prox.** We first demonstrate the communication efficiency of PEARL-Prox (using SGD subroutine) predicted by our theory. Corollary 4.8 implies that with $\gamma \equiv \frac{2\log\tau}{\lambda\tau}$, if $\tau$ is large enough, the performance of Algorithm 4 will be close to linear convergence. To verify this prediction, we run Algorithm 4 with theoretical parameter choice in Corollary 4.8 and plot the relative distance to the equilibrium $\frac{\|\boldsymbol{x}^p - \boldsymbol{x}^\star\|^2}{\|\boldsymbol{x}^0 - \boldsymbol{x}^\star\|^2}$ versus the number of communication $p$ (Figure 3), for varying $\tau$. We compare the cases $\tau \in \{17, 50, 100, 200\}$ (where $\tau = 17$ is the smallest possible value allowed under our parameter choice) with the case of exact proximal computation (dashed black line) where the convergence is linear (see Theorem 4.4). We indeed observe that with larger $\tau$, the performance plot becomes closer to that of exact PEARL-Prox, despite stochasticity.

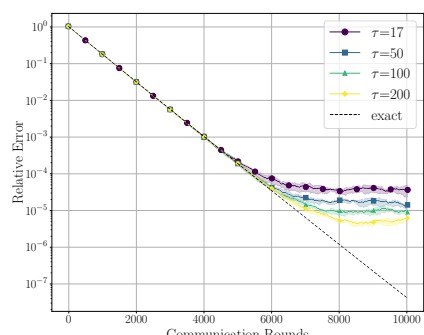

Figure 3: Plots of $\frac{\|\boldsymbol{x}^p - \boldsymbol{x}^\star\|^2}{\|\boldsymbol{x}^0 - \boldsymbol{x}^\star\|^2}$ for Algorithm 4 (PEARL-Prox with SGD) using theoretical parameters $\lambda = 4(\ell + L_{\max}\sqrt{\kappa})$, $\tau_i^p \equiv \tau$ and $\gamma_{i,t}^p \equiv \frac{2\log\tau}{\lambda\tau}$, for different $\tau$.

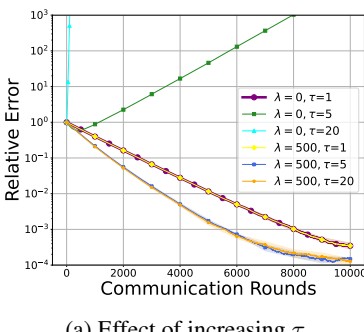 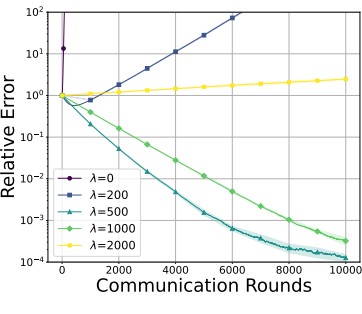

(a) Effect of increasing $\tau$            (b) Effect of changing $\lambda$

Figure 4: Plots of $\frac{\|x^p - x^\star\|^2}{\|x^0 - x^\star\|^2}$ for Algorithm 4 (PEARL-Prox with SGD) using fixed $\tau_i^p \equiv \tau$ and $\gamma_{i,t}^p \equiv \gamma$ for different values of $\lambda$. **(Left)** We fix $\gamma = 10^{-3}$ and test the effect of increasing $\tau$ with $\tau \in \{1, 5, 20\}$, for $\lambda = 0$ (PEARL-SGD) and $\lambda = 500$. **(Right)** We fix $\tau = 20$, $\gamma = 10^{-3}$ and test the effect of changing $\lambda$ with $\lambda \in \{0, 200, 500, 1000, 2000\}$.

**Player drift (effect of increasing $\tau$).** In the following experiment we illustrate that player drift occurs for PEARL-SGD but not for PEARL-Prox. We run Algorithm 4 with the fixed step size $\gamma_{i,t}^p \equiv \gamma = 10^{-3}$ and compare PEARL-SGD (recovered as the case $\lambda = 0$ from Algorithm 4) against PEARL-Prox with $\lambda = 500$ for varying values of $\tau_i^p \equiv \tau \in \{1, 5, 20\}$ (Figure 4a). We observe that PEARL-SGD only converges when $\tau = 1$ and diverges away to infinity with larger $\tau$, more quickly as $\tau$ increases. This indicates that the performance of PEARL-SGD is sensitive to $\tau$ (excessive local updates are not allowed), i.e., the player drift occurs, as predicted by the prior work Yoon et al. (2025) which suggested that $\gamma$ has to scale inversely proportionally with $\tau$. On the other hand, for PEARL-Prox, the proximal computation becomes more accurate as $\tau$ increases from 1 to 5, rather accelerating convergence. The convergence pattern does not change significantly as $\tau$ increases from 5 to 20 (or even larger) because the SGD subroutine already converges with $\tau = 5$ due to the large strong convexity parameter of the regularized objective ($\lambda = 500$), making it easy to optimize.

**$\lambda$ as a hyperparameter for convergence.** Here, we fix $\tau = 100$ and $\gamma = 10^{-3}$, and run Algorithm 4 with different $\lambda \in \{0, 200, 500, 1000, 2000\}$ (Figure 4b), demonstrating the usage of $\lambda$ as a hyperparameter capable of ensuring convergence in the setups where PEARL-SGD does not converge. PEARL-SGD ($\lambda = 0$) rapidly diverges because $\tau$ is too large for the given choice of $\gamma$. Divergence slows down as $\lambda$ increases, and at the sweet spot $\lambda = 500$ we observe the fastest convergence. However, increasing beyond $\lambda = 1000$ is suboptimal because then the curvature of the regularized objective grows beyond $\gamma^{-1}$ and the SGD subroutine may fail to converge (so one should keep $\lambda = \mathcal{O}(\gamma^{-1})$).

## 6 CONCLUSION

We provide a formal characterization of player drift in multiplayer federated learning, which indicates the divergence of game dynamics due to excessive local computations. As a resolution to player drift, we introduce the novel PEARL-Prox algorithm. We present theoretical convergence guarantees for PEARL-Prox, demonstrating that it can take the full advantage of large local computes, leading to a significant improvement in communication efficiency compared to PEARL-SGD, which suffers from player drift.

We foresee several possible directions of future work for resolving the current limitations: convergence analysis without (quasi) strong monotonicity, MpFL with model compression for cheaper communication and privacy, and algorithms for decentralized and asynchronous MpFL. While we believe that the current theory provides a solid understanding of the MpFL setup and communication efficiency in general, relaxing each layer of idealization required for the formal theory would be a valuable progression toward further practical impact.

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

# A    MORE COMPREHENSIVE DISCUSSION ON RELATED WORK

**Federated learning and heterogeneity.**    The standard formulation of federated learning (FL) is given by the finite sum minimization problem

$$\underset{x \in \mathbb{R}^d}{\text{minimze}} \; f(x) = \frac{1}{n} \sum_{i=1}^{n} f_i(x) \tag{7}$$

where $f_i$ is the local objective function of the $i$-th client and $x \in \mathbb{R}^d$ denotes the global model parameter to be optimized (McMahan et al., 2017; Kairouz et al., 2021). In most setups, one has $f_i(x) = \mathbb{E}_{\xi \sim \mathcal{D}_i}[\ell(\xi; x)]$ where $\ell(\xi; x)$ is the loss on the data $\xi$ (drawn from the local distribution $\mathcal{D}_i$) with the model parameter $x$ (McMahan et al., 2017; Li et al., 2020b; Khaled et al., 2020; Wang et al., 2020). A central challenge in FL is heterogeneity, which includes systems and statistical heterogeneity (Li et al., 2020a). Systems heterogeneity includes the discrepancy among clients in their capabilities to store, compute and communicate (Li et al., 2020a), as well as the issue of distinct clients being inactive or dropping out at each time (Bonawitz et al., 2019). Statistical heterogeneity refers to each client's data distribution being distinct (non-IID local datasets) (Konečný et al., 2016a; Smith et al., 2017; Li et al., 2020b). From the perspective of optimization, statistical heterogeneity manifests as differences in the local objectives $f_i(\cdot)$ and their gradients $\nabla f_i(\cdot)$, and several theoretical works establishing convergence of FL algorithms have made assumptions that the dissimilarity of these local gradients due to heterogeneity is bounded (Li et al., 2020b; Karimireddy et al., 2020; Gorbunov et al., 2021).

In MpFL, each player (client) has actions/models of distinct structures and dimensions ($x_i \in \mathbb{R}^{d_i}$ and $d_i$'s can be all different), and the local objective functions $f_i(x_i; x_{-i}) = \mathbb{E}_{\xi_i \sim \mathcal{D}_i}[f_{i,\xi_i}(x_i; x_{-i})]$ may differ in their fundamental structures (e.g., $f_{i,\xi_i}(\cdot)$ may have totally different forms for distinct $i$ rather than being based on some common loss function $\ell(\cdot)$) (Yoon et al., 2025). This essentially creates an additional dimension of heterogeneity arising from the game-theoretic formulation itself (other than systems or statistical heterogeneity). However, in this work, we do not formally distinguish the sources of heterogeneity and broadly refer to the distinction among $f_i$'s as "heterogeneity". Note that although we do not focus on systems heterogeneity in our work, the subroutine-agnostic nature of PEARL-Prox allows each player to flexibly optimize their regularized objective up to the accuracy level they can afford with their compute budget, offering a partial resolution to systems heterogeneity, as the popular FedProx algorithm (Li et al., 2020b) did for classical FL.

**Comparison with client drift and its algorithmic treatments.**    Performing local model updates in the FL setup (7) under (statistical) heterogeneity may cause the local optimization trajectories to diverge significantly, which biases the global model and even hinders its convergence (Konečný et al., 2016a; Zhao et al., 2018; Haddadpour & Mahdavi, 2019; Wang et al., 2020). This phenomenon is often referred to as client drift in the literature (Karimireddy et al., 2020; Koloskova et al., 2020; Gao et al., 2022; Grudzień et al., 2023), and multiple prior works have proposed algorithmic adjustments to FedAvg (Local SGD) as a resolution to this issue (Li et al., 2020b; Karimireddy et al., 2020; Gorbunov et al., 2021; Mitra et al., 2021; Mishchenko et al., 2022). Among them, we view the FedProx algorithm of Li et al. (2020b) most relevant to our PEARL-Prox, as FedProx also lets each client to minimize the regularized local objectives. However, a critical distinction is that FedProx averages the local models at synchronization (aggregation) steps, which is possible only in the classical FL setup and not in the MpFL scenario (note that the local actions $x_i$ may even have varying dimensions). In MpFL (and in PEARL-Prox), the central server collects and concatenates all local models and redistributes them to each player (Yoon et al., 2025). This distinction is analogous to the relationship between Local SGD and PEARL-SGD; despite formal similarities, their mechanisms are fundamentally different. For a more direct comparison between the client drift and player drift phenomena, we refer the readers to Yoon et al. (2025, Section 3.2).

## B  MISSING PROOFS FOR SECTION 4

### B.1  PROOF OF LEMMA 4.6

Note that $\text{Prox}_{\frac{1}{\lambda}f_i(\cdot;x_{-i}^p)}(x_i^p) = \left(\text{Id} + \frac{1}{\lambda}\nabla_{x_i}f_i(\cdot;x_{-i}^p)\right)^{-1}(x_i^p)$ and

$$\nabla_{x_i}f_i(\cdot;x_{-i}^p)\colon \mathbb{R}^{d_i} \to \mathbb{R}^{d_i}$$

is a monotone operator, so $\text{Prox}_{\frac{1}{\lambda}f_i(\cdot;x_{-i}^p)}\colon \mathbb{R}^{d_i} \to \mathbb{R}^{d_i}$ is nonexpansive. Next, we have

$$x_i^p = \text{Prox}_{\frac{1}{\lambda}f_i(\cdot;x_{-i}^p)}\left(x_i^p + \frac{1}{\lambda}\nabla_{x_i}f_i(x_i^p;x_{-i}^p)\right)$$

and therefore,

$$
\begin{aligned}
\left\|x_i^p - \text{Prox}_{\frac{1}{\lambda}f_i(\cdot;x_{-i}^p)}(x_i^p)\right\| &= \left\|\text{Prox}_{\frac{1}{\lambda}f_i(\cdot;x_{-i}^p)}\left(x_i^p + \frac{1}{\lambda}\nabla_{x_i}f_i(x_i^p;x_{-i}^p)\right) - \text{Prox}_{\frac{1}{\lambda}f_i(\cdot;x_{-i}^p)}(x_i^p)\right\| \\
&\le \left\|\left(x_i^p + \frac{1}{\lambda}\nabla_{x_i}f_i(x_i^p;x_{-i}^p)\right) - x_i^p\right\| \\
&= \frac{1}{\lambda}\left\|\nabla_{x_i}f_i(x_i^p;x_{-i}^p)\right\|.
\end{aligned}
$$

### B.2  PROOF OF THEOREM 4.4

As each player computes proximal operator exactly, we have

$$x_i^{p+1} = \text{Prox}_{\frac{1}{\lambda}f_i(\cdot;x_{-i}^p)}(x_i^p)$$

for $i = 1, \ldots, n$ and therefore, by Lemma 4.6,

$$\left\|x_i^p - x_i^{p+1}\right\| \le \frac{1}{\lambda}\left\|\nabla_{x_i}f_i(x_i^p;x_{-i}^p)\right\|.$$

Now observe that for each $p = 0, \ldots, R-1$,

$$
\begin{aligned}
\left\|\boldsymbol{x}^{p+1} - \boldsymbol{x}^\star\right\|^2 &= \left\|\boldsymbol{x}^p - \boldsymbol{x}^\star - (\boldsymbol{x}^p - \boldsymbol{x}^{p+1})\right\|^2 \\
&= \left\|\boldsymbol{x}^p - \boldsymbol{x}^\star\right\|^2 - 2\left\langle \boldsymbol{x}^p - \boldsymbol{x}^{p+1}, \boldsymbol{x}^p - \boldsymbol{x}^\star\right\rangle + \left\|\boldsymbol{x}^p - \boldsymbol{x}^{p+1}\right\|^2 \\
&= \left\|\boldsymbol{x}^p - \boldsymbol{x}^\star\right\|^2 - 2\sum_{i=1}^n \left\langle x_i^p - x_i^{p+1}, x_i^p - x_i^\star\right\rangle + \sum_{i=1}^n \left\|x_i^p - x_i^{p+1}\right\|^2.
\end{aligned}
\tag{8}
$$

Because $x_i^p - x_i^{p+1} = \frac{1}{\lambda}\nabla_{x_i}f_i(x_i^{p+1};x_{-i}^p)$, we can write

$$x_i^p - x_i^{p+1} = \frac{1}{\lambda}\nabla_{x_i}f_i(x_i^p;x_{-i}^p) + \frac{1}{\lambda}\underbrace{\left(\nabla_{x_i}f_i(x_i^{p+1};x_{-i}^p) - \nabla_{x_i}f_i(x_i^p;x_{-i}^p)\right)}_{:=\delta_i^p}$$

where $\|\delta_i^p\| \leq L_i \left\| x_i^{p+1} - x_i^p \right\|$. Plugging this into (8) we obtain

$$\left\| \boldsymbol{x}^{p+1} - \boldsymbol{x}^\star \right\|^2 = \left\| \boldsymbol{x}^p - \boldsymbol{x}^\star \right\|^2 - \frac{2}{\lambda} \sum_{i=1}^n \left\langle \nabla_{x_i} f_i(x_i^p; x_{-i}^p), x_i^p - x_i^\star \right\rangle$$

$$- \frac{2}{\lambda} \sum_{i=1}^n \left\langle \delta_i^p, x_i^p - x_i^\star \right\rangle + \sum_{i=1}^n \left\| x_i^p - x_i^{p+1} \right\|^2$$

$$\leq \left\| \boldsymbol{x}^p - \boldsymbol{x}^\star \right\|^2 - \frac{2}{\lambda} \left\langle F(\boldsymbol{x}^p), \boldsymbol{x}^p - \boldsymbol{x}^\star \right\rangle$$

$$+ \frac{1}{\lambda} \sum_{i=1}^n \left( \alpha \left\| x_i^p - x_i^\star \right\|^2 + \frac{1}{\alpha} \left\| \delta_i^p \right\|^2 \right) + \sum_{i=1}^n \left\| x_i^p - x_i^{p+1} \right\|^2$$

$$\leq \left( 1 + \frac{\alpha}{\lambda} \right) \left\| \boldsymbol{x}^p - \boldsymbol{x}^\star \right\|^2 - \frac{2}{\lambda} \left\langle F(\boldsymbol{x}^p), \boldsymbol{x}^p - \boldsymbol{x}^\star \right\rangle + \sum_{i=1}^n \left( 1 + \frac{L_i^2}{\lambda\alpha} \right) \left\| x_i^p - x_i^{p+1} \right\|^2$$

$$\leq \left( 1 + \frac{\alpha}{\lambda} \right) \left\| \boldsymbol{x}^p - \boldsymbol{x}^\star \right\|^2 - \frac{2}{\lambda} \left\langle F(\boldsymbol{x}^p), \boldsymbol{x}^p - \boldsymbol{x}^\star \right\rangle + \left( 1 + \frac{L_{\max}^2}{\lambda\alpha} \right) \sum_{i=1}^n \frac{1}{\lambda^2} \left\| \nabla_{x_i} f_i(x_i^p; x_{-i}^p) \right\|^2$$

$$= \left( 1 + \frac{\alpha}{\lambda} \right) \left\| \boldsymbol{x}^p - \boldsymbol{x}^\star \right\|^2 - \frac{2}{\lambda} \left\langle F(\boldsymbol{x}^p), \boldsymbol{x}^p - \boldsymbol{x}^\star \right\rangle + \left( 1 + \frac{L_{\max}^2}{\lambda\alpha} \right) \frac{1}{\lambda^2} \left\| F(\boldsymbol{x}^p) \right\|^2$$

where for the first inequality, we use Young's inequality (with $\alpha > 0$ to be determined later), for the second inequality we use the identity $\|\boldsymbol{x}^p - \boldsymbol{x}^\star\|^2 = \sum_{i=1}^n \|x_i^p - x_i^\star\|^2$ and $\|\delta_i^p\| \leq L_i \left\| x_i^{p+1} - x_i^p \right\|$, and for the last inequality, we use Lemma 4.6 with $L_i \leq L_{\max}$.

Now, using star-cocoercivity of $F$ we have

$$\left\| \boldsymbol{x}^{p+1} - \boldsymbol{x}^\star \right\|^2 \leq \left( 1 + \frac{\alpha}{\lambda} \right) \left\| \boldsymbol{x}^p - \boldsymbol{x}^\star \right\|^2 - \frac{2}{\lambda} \left\langle F(\boldsymbol{x}^p), \boldsymbol{x}^p - \boldsymbol{x}^\star \right\rangle + \left( 1 + \frac{L_{\max}^2}{\lambda\alpha} \right) \frac{\ell}{\lambda^2} \left\langle F(\boldsymbol{x}^p), \boldsymbol{x}^p - \boldsymbol{x}^\star \right\rangle$$

$$\leq \left( 1 + \frac{\alpha}{\lambda} \right) \left\| \boldsymbol{x}^p - \boldsymbol{x}^\star \right\|^2 - \underbrace{\left( \frac{2}{\lambda} - \left( 1 + \frac{L_{\max}^2}{\lambda\alpha} \right) \frac{\ell}{\lambda^2} \right)}_{:=C} \left\langle F(\boldsymbol{x}^p), \boldsymbol{x}^p - \boldsymbol{x}^\star \right\rangle.$$

Provided that $C > 0$, we can further use $\mu$-quasi-strong monotonicity of $F$ to obtain

$$\left\| \boldsymbol{x}^{p+1} - \boldsymbol{x}^\star \right\|^2 \leq \left( 1 + \frac{\alpha}{\lambda} \right) \left\| \boldsymbol{x}^p - \boldsymbol{x}^\star \right\|^2 - C\mu \left\| \boldsymbol{x}^p - \boldsymbol{x}^\star \right\|^2$$

$$= \left[ 1 + \frac{\alpha}{\lambda} - \mu \left( \frac{2}{\lambda} - \frac{\ell}{\lambda^2} \left( 1 + \frac{L_{\max}^2}{\lambda\alpha} \right) \right) \right] \left\| \boldsymbol{x}^p - \boldsymbol{x}^\star \right\|^2.$$

Now to minimize the last factor

$$1 + \frac{\alpha}{\lambda} - \mu \left( \frac{2}{\lambda} - \frac{\ell}{\lambda^2} \left( 1 + \frac{L_{\max}^2}{\lambda\alpha} \right) \right) = 1 - \frac{2\mu}{\lambda} + \frac{\mu\ell}{\lambda^2} + \frac{\alpha}{\lambda} + \frac{\mu\ell L_{\max}^2}{\lambda^3 \alpha}$$

with respect to $\alpha$, we choose $\alpha = \frac{L_{\max}\sqrt{\mu\ell}}{\lambda}$, which gives

$$1 - \frac{2\mu}{\lambda} + \frac{\mu\ell}{\lambda^2} + \frac{2L_{\max}\sqrt{\mu\ell}}{\lambda^2} = 1 - \frac{2\mu}{\lambda} \left( 1 - \frac{\ell}{2\lambda} - \frac{L_{\max}\sqrt{\ell/\mu}}{\lambda} \right) = 1 - \frac{2\mu\zeta}{\lambda}$$

where $\zeta = 1 - \frac{\ell + 2L_{\max}\sqrt{\kappa}}{2\lambda} > 0$ by the choice of $\lambda$. Finally, we verify that with our choice of $\alpha$,

$$C = \frac{2}{\lambda} \left( 1 - \frac{\ell + L_{\max}\sqrt{\kappa}}{2\lambda} \right) > \frac{2}{\lambda} \left( \zeta + \frac{L_{\max}\sqrt{\kappa}}{2\lambda} \right) > 0.$$

### B.3 Proof of Theorem 4.7

Denote $\tilde{x}_i^{p+1} = \text{Prox}_{\frac{1}{\lambda} f_i(\cdot; x_{-i}^p)}(x_i^p)$, so that $x_i^{p+1} = \tilde{x}_i^{p+1} + v_i^p$ where $v_i^p \in \mathbb{R}^{d_i}$ is a random vector such that and $\mathbb{E}\left[\|v_i^p\|^2 \Big| x_i^p\right] \le \delta_i^p \left\|\nabla_{x_i} f_i(x_i^p; x_{-i}^p)\right\|^2 + \epsilon_i^p$. Then for each $i = 1, \dots, n$, we have

$$
\begin{aligned}
\mathbb{E}\left[\left\|x_i^p - x_i^{p+1}\right\|^2 \Big| x_i^p\right] &= \mathbb{E}\left[\left\|x_i^p - \tilde{x}_i^{p+1} - v_i^p\right\|^2 \Big| x_i^p\right] \\
&\le \mathbb{E}\left[2\left\|x_i^p - \tilde{x}_i^{p+1}\right\|^2 + 2\|v_i^p\|^2 \Big| x_i^p\right] \\
&= 2\left\|x_i^p - \tilde{x}_i^{p+1}\right\|^2 + 2\left(\delta_i^p \left\|\nabla_{x_i} f_i(x_i^p; x_{-i}^p)\right\|^2 + \epsilon_i^p\right) \quad (9)
\end{aligned}
$$

where in the third line, we use the fact that $\tilde{x}_i^{p+1}$ is a non-random quantity given $x_i^p$. Now observe that for each $p = 0, \dots, R-1$,

$$
\begin{aligned}
\left\|\boldsymbol{x}^{p+1} - \boldsymbol{x}^\star\right\|^2 &= \left\|\boldsymbol{x}^p - \boldsymbol{x}^\star - (\boldsymbol{x}^p - \boldsymbol{x}^{p+1})\right\|^2 \\
&= \|\boldsymbol{x}^p - \boldsymbol{x}^\star\|^2 - 2\left\langle \boldsymbol{x}^p - \boldsymbol{x}^{p+1}, \boldsymbol{x}^p - \boldsymbol{x}^\star\right\rangle + \left\|\boldsymbol{x}^p - \boldsymbol{x}^{p+1}\right\|^2 \\
&= \|\boldsymbol{x}^p - \boldsymbol{x}^\star\|^2 - 2\sum_{i=1}^n \left\langle x_i^p - x_i^{p+1}, x_i^p - x_i^\star\right\rangle + \sum_{i=1}^n \left\|x_i^p - x_i^{p+1}\right\|^2. \quad (10)
\end{aligned}
$$

Because $x_i^p - x_i^{p+1} = x_i^p - \tilde{x}_i^{p+1} - v_i^p = \frac{1}{\lambda}\nabla_{x_i} f_i(\tilde{x}_i^{p+1}; x_{-i}^p) - v_i^p$, we can write

$$
x_i^p - x_i^{p+1} = \frac{1}{\lambda}\nabla_{x_i} f_i(x_i^p; x_{-i}^p) + \frac{1}{\lambda}\left(\nabla_{x_i} f_i(\tilde{x}_i^{p+1}; x_{-i}^p) - \nabla_{x_i} f_i(x_i^p; x_{-i}^p)\right) - v_i^p.
$$

Plugging this into the inner product term of (10), we obtain

$$
\begin{aligned}
\left\|\boldsymbol{x}^{p+1} - \boldsymbol{x}^\star\right\|^2 &= \|\boldsymbol{x}^p - \boldsymbol{x}^\star\|^2 - \frac{2}{\lambda}\sum_{i=1}^n \left\langle \nabla_{x_i} f_i(x_i^p; x_{-i}^p), x_i^p - x_i^\star\right\rangle \\
&\quad - 2\sum_{i=1}^n \left\langle \frac{1}{\lambda}\left(\nabla_{x_i} f_i(\tilde{x}_i^{p+1}; x_{-i}^p) - \nabla_{x_i} f_i(x_i^p; x_{-i}^p)\right) - v_i^p, x_i^p - x_i^\star\right\rangle + \sum_{i=1}^n \left\|x_i^p - x_i^{p+1}\right\|^2 \\
&\le \|\boldsymbol{x}^p - \boldsymbol{x}^\star\|^2 - \frac{2}{\lambda}\left\langle F(\boldsymbol{x}^p), \boldsymbol{x}^p - \boldsymbol{x}^\star\right\rangle \\
&\quad + \frac{1}{\lambda}\sum_{i=1}^n \left(\alpha\|x_i^p - x_i^\star\|^2 + \frac{1}{\alpha}\left\|\nabla_{x_i} f_i(\tilde{x}_i^{p+1}; x_{-i}^p) - \nabla_{x_i} f_i(x_i^p; x_{-i}^p)\right\|^2\right) \\
&\quad + \sum_{i=1}^n \left(\beta\|x_i^p - x_i^\star\|^2 + \frac{1}{\beta}\|v_i^p\|^2\right) + \sum_{i=1}^n \left\|x_i^p - x_i^{p+1}\right\|^2 \\
&\le \left(1 + \frac{\alpha}{\lambda} + \beta\right)\|\boldsymbol{x}^p - \boldsymbol{x}^\star\|^2 - \frac{2}{\lambda}\left\langle F(\boldsymbol{x}^p), \boldsymbol{x}^p - \boldsymbol{x}^\star\right\rangle + \sum_{i=1}^n \frac{L_i^2}{\lambda\alpha}\left\|\tilde{x}_i^{p+1} - x_i^p\right\|^2 \\
&\quad + \sum_{i=1}^n \left\|x_i^p - x_i^{p+1}\right\|^2 + \frac{1}{\beta}\sum_{i=1}^n \|v_i^p\|^2
\end{aligned}
$$

where the last inequality uses $L_i$-smoothness of $f_i(\cdot; x_{-i}^p)$ and the constants $\alpha, \beta > 0$ arising from the application of Young's inequality are to be determined later. Taking the expectation of both sides (conditioned on $\boldsymbol{x}^p$) and using (9), we obtain

$$
\begin{aligned}
\mathbb{E}\left[\|\boldsymbol{x}^{p+1} - \boldsymbol{x}^\star\|^2 \Big| \boldsymbol{x}^p\right] &\le \left(1 + \frac{\alpha}{\lambda} + \beta\right)\|\boldsymbol{x}^p - \boldsymbol{x}^\star\|^2 - \frac{2}{\lambda}\left\langle F(\boldsymbol{x}^p), \boldsymbol{x}^p - \boldsymbol{x}^\star\right\rangle \\
&\quad + \sum_{i=1}^n \left(2 + \frac{L_i^2}{\lambda\alpha}\right)\left\|x_i^p - \tilde{x}_i^{p+1}\right\|^2 + \sum_{i=1}^n 2\delta_i^p\left\|\nabla_{x_i} f_i(x_i^p; x_{-i}^p)\right\|^2 + \left(2 + \frac{1}{\beta}\right)\sum_{i=1}^p \epsilon_i^p.
\end{aligned}
$$

Now using Lemma 4.6 to bound $\left\| x_i^p - \tilde{x}_i^{p+1} \right\|^2 \leq \frac{1}{\lambda^2} \left\| \nabla_{x_i} f_i(x_i^p; x_{-i}^p) \right\|^2$ and denoting $\delta_{\max} = \max\limits_{\substack{i=1,\dots,n \\ p=0,\dots,R-1}} \delta_i^p$ we obtain

$$
\mathbb{E}\left[ \left\| \boldsymbol{x}^{p+1} - \boldsymbol{x}^\star \right\|^2 \Big| \boldsymbol{x}^p \right] \leq \left(1 + \frac{\alpha}{\lambda} + \beta \right) \left\| \boldsymbol{x}^p - \boldsymbol{x}^\star \right\|^2 - \frac{2}{\lambda} \left\langle F(\boldsymbol{x}^p), \boldsymbol{x}^p - \boldsymbol{x}^\star \right\rangle
$$
$$
+ \sum_{i=1}^n \left( \frac{1}{\lambda^2} \left( 2 + \frac{L_i^2}{\lambda\alpha} \right) + 2\delta_i^p \right) \left\| \nabla_{x_i} f_i(x_i^p; x_{-i}^p) \right\|^2 + \left( 2 + \frac{1}{\beta} \right) \sum_{i=1}^p \epsilon_i^p
$$
$$
\leq \left(1 + \frac{\alpha}{\lambda} + \beta \right) \left\| \boldsymbol{x}^p - \boldsymbol{x}^\star \right\|^2 - \frac{2}{\lambda} \left\langle F(\boldsymbol{x}^p), \boldsymbol{x}^p - \boldsymbol{x}^\star \right\rangle
$$
$$
+ \left[ \frac{1}{\lambda^2} \left( 2 + \frac{L_{\max}^2}{\lambda\alpha} \right) + 2\delta_{\max} \right] \left\| F(\boldsymbol{x}^p) \right\|^2 + \left( 2 + \frac{1}{\beta} \right) \sum_{i=1}^n \epsilon_i^p
$$
$$
\leq \left(1 + \frac{\alpha}{\lambda} + \beta \right) \left\| \boldsymbol{x}^p - \boldsymbol{x}^\star \right\|^2 - \left[ \frac{2}{\lambda} - \ell \left( \frac{1}{\lambda^2} \left( 2 + \frac{L_{\max}^2}{\lambda\alpha} \right) + 2\delta_{\max} \right) \right] \left\langle F(\boldsymbol{x}^p), \boldsymbol{x}^p - \boldsymbol{x}^\star \right\rangle
$$
$$
+ \left( 2 + \frac{1}{\beta} \right) \sum_{i=1}^n \epsilon_i^p
$$

where the last inequality uses star-cocoercivity $\| F(\boldsymbol{x}^p) \|^2 \leq \ell \left\langle F(\boldsymbol{x}^p), \boldsymbol{x}^p - \boldsymbol{x}^\star \right\rangle$. Next, assuming $C = \frac{2}{\lambda} - \frac{1}{\lambda^2}\left( 2 + \frac{L_{\max}^2}{\lambda\alpha} \right) - 2\delta_{\max} > 0$ (we verify this below with a particular choice of $\alpha$) and using quasi-strong monotonicity $\langle F(\boldsymbol{x}^p), \boldsymbol{x}^p - \boldsymbol{x}^\star \rangle \geq \mu \| \boldsymbol{x}^p - \boldsymbol{x}^\star \|^2$, we have

$$
\mathbb{E}\left[ \left\| \boldsymbol{x}^{p+1} - \boldsymbol{x}^\star \right\|^2 \Big| \boldsymbol{x}^p \right]
$$
$$
\leq \left[ 1 + \frac{\alpha}{\lambda} + \beta - \frac{2\mu}{\lambda} + \mu\ell \left( \frac{1}{\lambda^2} \left( 2 + \frac{L_{\max}^2}{\lambda\alpha} \right) + 2\delta_{\max} \right) \right] \left\| \boldsymbol{x}^p - \boldsymbol{x}^\star \right\|^2 + \left( 2 + \frac{1}{\beta} \right) \sum_{i=1}^n \epsilon_i^p \quad (11)
$$

Now we take $\beta = \frac{\mu}{2\lambda}$ and use the condition $\delta_{\max} \leq \frac{1}{4\lambda\ell} \implies 2\mu\ell\delta_{\max} \leq \frac{\mu}{2\lambda}$ to bound the coefficient of the $\| \boldsymbol{x}^p - \boldsymbol{x}^\star \|^2$ term as

$$
1 + \frac{\alpha}{\lambda} + \beta - \frac{2\mu}{\lambda} + \mu\ell \left( \frac{1}{\lambda^2} \left( 2 + \frac{L_{\max}^2}{\lambda\alpha} \right) + 2\delta_{\max} \right) \leq 1 - \frac{\mu}{\lambda} + \frac{2\mu\ell}{\lambda^2} + \frac{\alpha}{\lambda} + \frac{\mu\ell L_{\max}^2}{\lambda^3\alpha}.
$$

Now taking $\alpha = \frac{L_{\max}\sqrt{\mu\ell}}{\lambda}$ to optimize the right hand side, (11) becomes

$$
\mathbb{E}\left[ \left\| \boldsymbol{x}^{p+1} - \boldsymbol{x}^\star \right\|^2 \Big| \boldsymbol{x}^p \right] \leq \left( 1 - \frac{\mu}{\lambda} + \frac{2\mu\ell + 2\sqrt{\mu\ell}L_{\max}}{\lambda^2} \right) \left\| \boldsymbol{x}^p - \boldsymbol{x}^\star \right\|^2 + \left( 2 + \frac{2\lambda}{\mu} \right) \sum_{i=1}^n \epsilon_i^p
$$
$$
\leq \left( 1 - \frac{\mu\zeta}{\lambda} \right) \left\| \boldsymbol{x}^p - \boldsymbol{x}^\star \right\|^2 + \left( 2 + \frac{2\lambda}{\mu} \right) \sum_{i=1}^n \epsilon_i^p
$$

where $\zeta = 1 - \frac{2\ell + 2L_{\max}\sqrt{\kappa}}{\lambda} > 0$. Note that with this choice of $\alpha$, we have $1 + \frac{\alpha}{\lambda} + \beta - C\mu = 1 - \frac{\mu\zeta}{\lambda} < 1$, which implies that $C > 0$, so the step deriving (11) is valid. Unrolling the recursion and taking the total expectation, we conclude that

$$
\mathbb{E}\left[ \left\| \boldsymbol{x}^R - \boldsymbol{x}^\star \right\|^2 \right] \leq \left( 1 - \frac{\mu\zeta}{\lambda} \right)^R \left\| \boldsymbol{x}^0 - \boldsymbol{x}^\star \right\|^2 + \left( 2 + \frac{2\lambda}{\mu} \right) \sum_{p=0}^{R-1} \left( 1 - \frac{\mu\zeta}{\lambda} \right)^{R-p-1} \sum_{i=1}^n \epsilon_i^p.
$$

### B.4 PROOF OF COROLLARY 4.8

We first state a general convergence result for SGD.

**Lemma B.1.** Let $h(\cdot) = \mathbb{E}_{\xi \sim \mathcal{D}}[h_\xi(\cdot)] \colon \mathbb{R}^d \to \mathbb{R}$ be $\mu$-strongly convex and $L$-smooth, and assume that there exists $\sigma > 0$ such that $\mathbb{E}\left[\|\nabla h_\xi(x) - \nabla h(x)\|^2\right] \leq \sigma^2$ for all $x \in \mathbb{R}^d$. Then SGD with constant step size $\gamma \in (0, \frac{1}{2L}]$ satisfies

$$\left\|x^T - x^\star\right\|^2 \leq (1 - \gamma\mu)^T \left\|x^0 - x^\star\right\|^2 + \frac{\gamma\sigma^2}{\mu}.$$

The proof is standard and can be found in, e.g., Stich (2019) (see Lemma 1 and the discussion following it).

Now, we apply Lemma B.1 to the regularized objectives $h(\cdot) = f_i(\cdot; x^p_{-i}) + \frac{\lambda}{2} \|\cdot - x^p_i\|^2$ (which are $\lambda$-strongly convex, $(\lambda + L_i)$-smooth) with unique minimum $\text{Prox}_{\frac{1}{\lambda} f_i(\cdot; x^p_{-i})}(x^p_i)$, and $\nabla_{x_i} f_{i, \xi_i}(\cdot; x^p_{-i})$ which are unbiased estimators of $\nabla_{x_i} f_i(\cdot; x^p_{-i})$ with bounded variance $\leq \sigma_i^2$ (by Assumption 4.3). Then, provided that $\gamma \leq \frac{1}{2(\lambda + L_{\max})}$, Algorithm 4 with $\tau_i^p \equiv \tau$ using constant step size $\gamma$ satisfies

$$\left\|x_i^{p+1} - \text{Prox}_{\frac{1}{\lambda} f_i(\cdot; x^p_{-i})}(x_i^p)\right\|^2 \leq (1 - \gamma\lambda)^\tau \left\|x_i^p - \text{Prox}_{\frac{1}{\lambda} f_i(\cdot; x^p_{-i})}(x_i^p)\right\|^2 + \frac{\gamma\sigma_i^2}{\lambda}$$

$$\leq (1 - \gamma\lambda)^\tau \frac{1}{\lambda^2} \left\|\nabla_{x_i} f_i(x_i^p; x^p_{-i})\right\|^2 + \frac{\gamma\sigma_i^2}{\lambda}$$

$$\leq \frac{e^{-\gamma\lambda\tau}}{\lambda^2} \left\|\nabla_{x_i} f_i(x_i^p; x^p_{-i})\right\|^2 + \frac{\gamma\sigma_i^2}{\lambda}.$$

where in the second line we use Lemma 4.6 and the third line uses $1 + t \leq e^t$ for $t \in \mathbb{R}$. Now taking $\gamma = \frac{2 \log \tau}{\lambda\tau}$ the above bound becomes

$$\left\|x_i^{p+1} - \text{Prox}_{\frac{1}{\lambda} f_i(\cdot; x^p_{-i})}(x_i^p)\right\|^2 \leq \frac{1}{\lambda^2\tau^2} \left\|\nabla_{x_i} f_i(x_i^p; x^p_{-i})\right\|^2 + \frac{2\sigma_i^2 \log \tau}{\lambda^2\tau}$$

Hence, Algorithm 4 satisfies Assumption 4.5 with $\delta_i^p = \frac{1}{\lambda^2\tau^2}$ and $\epsilon_i^p = \frac{2\sigma_i^2 \log \tau}{\lambda^2\tau}$. Assuming that $\frac{1}{\lambda^2\tau^2} \leq \frac{1}{4\lambda\ell}$ we can apply Theorem 4.7, which implies

$$\mathbb{E}\left[\left\|\boldsymbol{x}^R - \boldsymbol{x}^\star\right\|^2\right] \leq \left(1 - \frac{\mu\zeta}{\lambda}\right)^R \left\|\boldsymbol{x}^0 - \boldsymbol{x}^\star\right\|^2 + \left(2 + \frac{2\lambda}{\mu}\right) \sum_{p=0}^{R-1} \left(1 - \frac{\mu\zeta}{\lambda}\right)^{R-p-1} \sum_{i=1}^n \epsilon_i^p$$

$$\leq \left(1 - \frac{\mu\zeta}{\lambda}\right)^R \left\|\boldsymbol{x}^0 - \boldsymbol{x}^\star\right\|^2 + \left(2 + \frac{2\lambda}{\mu}\right) \frac{2 \log \tau}{\lambda^2\tau} \sum_{q=0}^\infty \left(1 - \frac{\mu\zeta}{\lambda}\right)^q \sum_{i=1}^n \sigma_i^2$$

$$= \left(1 - \frac{\mu\zeta}{\lambda}\right)^R \left\|\boldsymbol{x}^0 - \boldsymbol{x}^\star\right\|^2 + \left(2 + \frac{2\lambda}{\mu}\right) \frac{2 \log \tau}{\lambda^2\tau} \sum_{q=0}^\infty \left(1 - \frac{\mu\zeta}{\lambda}\right)^q \sum_{i=1}^n \sigma_i^2$$

$$= \left(1 - \frac{\mu\zeta}{\lambda}\right)^R \left\|\boldsymbol{x}^0 - \boldsymbol{x}^\star\right\|^2 + \left(2 + \frac{2\lambda}{\mu}\right) \frac{2\sigma^2 \log \tau}{\mu\zeta\lambda\tau}.$$

Finally, note that for the above to hold true we need to verify the two conditions on $\tau$: first, $\delta_{\max} = \frac{1}{\lambda^2\tau^2} \leq \frac{1}{4\lambda\ell} \iff \tau \geq \sqrt{\frac{4\ell}{\lambda}}$ and $\gamma = \frac{2 \log \tau}{\lambda\tau} \leq \frac{1}{2(\lambda + L_{\max})}$. The second condition is satisfied if $\tau \geq 16 \left(1 + \frac{L_{\max}}{\lambda}\right)^2$, as this implies

$$\frac{2 \log \tau}{\lambda\tau} \leq \frac{2}{\lambda\sqrt{\tau}} \leq \frac{2}{4\lambda \left(1 + \frac{L_{\max}}{\lambda}\right)} = \frac{1}{2(\lambda + L_{\max})}$$

where we use $\log \tau \leq \sqrt{\tau}$ for $\tau \geq 1$.

## C  Discussion on communication complexity

In Yoon et al. (2025, Corollary 3.5, Theorem 3.6), it is shown that PEARL-SGD (with properly tuned hyperparameters) converges with the rate of

$$\mathbb{E}\left[\left\|\boldsymbol{x}^T - \boldsymbol{x}^\star\right\|^2\right] = \tilde{\mathcal{O}}\left(\frac{\left\|\boldsymbol{x}^0 - \boldsymbol{x}^\star\right\|^2}{T^2} + \frac{\sigma^2}{\mu^2 T} + \frac{\tau^2 \sigma^2}{\mu^3 T^2}\right) \tag{12}$$

where $\tau$ is the number of local SGD steps performed at each round, $R$ is the number of communication rounds and $T = \tau R$ is the total number of iterations. (Note that here the iterates are numbered differently from our notation; their index corresponds to the cumulative number of gradient computations performed to produce it starting from $\boldsymbol{x}^0$, while we only keep track of the number of communications performed.) The bound (12) implies that with $\tau = \Theta(\sqrt{T})$, one can achieve a convergence rate that is asymptotically not slower than the fully communicating case $\tau = 1$, while requiring a reduced number of communications $R = T/\tau = \Theta(\sqrt{T})$. In this case, the number of communications required to achieve $\mathbb{E}\left[\left\|\boldsymbol{x}^T - \boldsymbol{x}^\star\right\|^2\right] \leq \epsilon$ is $\Theta(\epsilon^{-1/2})$. The communication cost cannot be reduced further by increasing $\tau$ beyond $\Theta(\sqrt{T})$ because then the last term in (12), proportional to $\tau^2/T^2 = 1/R^2$, becomes dominant, and one will still require $R = \Omega(\epsilon^{-1/2})$ to attain $\mathbb{E}\left[\left\|\boldsymbol{x}^T - \boldsymbol{x}^\star\right\|^2\right] \leq \epsilon$.

On the other hand, in our Corollary 4.8 we show

$$\mathbb{E}\left[\left\|\boldsymbol{x}^R - \boldsymbol{x}^\star\right\|^2\right] = \mathcal{O}\left(e^{-\mu \zeta R/\lambda}\left\|\boldsymbol{x}^0 - \boldsymbol{x}^\star\right\|^2 + \frac{\sigma^2 \log \tau}{\mu^2 \tau}\right),$$

which implies that the dominant term (proportional to $\log \tau/\tau$) is independent of $R$ and decreases in $\tau$, unlike (12). This is why PEARL-Prox can take full advantage of large $\tau$ (the convergence bound only gets improved in the limit $\tau \to \infty$), and in principle, can achieve $\mathbb{E}\left[\left\|\boldsymbol{x}^R - \boldsymbol{x}^\star\right\|^2\right] \leq \epsilon$ using $R = \Theta(\log \epsilon^{-1})$ communications, provided that $\tau$ is large enough so that $\frac{\log \tau}{\tau} = \mathcal{O}\left(\frac{\mu^2 \epsilon}{\sigma^2}\right)$.

