# OpenReview forum: "PEARL-Prox: Proximal Algorithm for Resolving Player Drift in Multiplayer Federated Learning"
_ICLR.cc/2026/Conference — Submitted to ICLR 2026_

### Official Review · Reviewer_SqrS · 2025-10-16

**Soundness:** 2
**Presentation:** 3
**Contribution:** 2
**Rating:** 2
**Confidence:** 4

**Summary:**

The paper introduces an algorithm named PEARL-Prox which is the proximal counterpart of the PEARL-SGD algorithm used in the multiplayer federated learning problem. The authors define the conception of player drift and analyze the proposed algorithm in the exact / inexact setting under assumptions such as smoothness and monotonicity and provide the corresponding convergence guarantee. The obtained communication complexity is compared to that of the PEARL-SGD. Numerical experiments are provided to validate the claims made in the paper.

**Strengths:**

(i) The paper overall is well-written and the logic is easy to follow.

(ii) The authors introduce and analyze the PEARL-Prox algorithm, which is novel, convergence guarantees are provided with theoretical intuitions.

(iii) The concept of player drift is introduced to help better understand the convergence in this case.

**Weaknesses:**

(i) The claimed better communication complexity of the PEARL-Prox algorithm seems to rely on selecting $\lambda$ optimally. However, the selection of $\lambda$ depends on $L_{\max}$ and the condition number $\kappa$, which is hard or impossible to compute. This also applies to Corollary 4.8 when we are use SGD as a subroutine, and we need to select the local step size. In addition, the comparison happens when $\tau$ (the local steps) goes to infinity, which is not practical. Therefore I do not believe that the comparison between those complexities are well justified. I believe this limitation is critical because the major contribution of the paper ties to the claim of improved complexity.

(ii) The proposed algorithm is novel only in the sense that it is applied in the MpFL setting, which is mainly theoretical. It is generally well known that the proximal framework can be used to mitigate the drift in FL, and compared to FedAvg or gradient based algorithms, we are essentially adding a regularizer to control how far each client can proceed in each step. In this sense, it is not unexpected that switching from PEARL-SGD to Prox yields such benefits, given the previous examples such as FedProx, ProxSkip and so on.

(iii) I do not get the motivation why we are considering MpFL setting, it seems to be very related to the Personalized Federated Learning framework where proximal algorithms are known to work very well. Is there a real world problem that the setting applies to while simpler frameworks can not? It seems to me that only unnecessary mathematical complications are added to the framework, but the practical problems can already be represented by simper framework.

(iv) The empirical evidence seems to be very limited. I understand that the authors mentioned that right now the MpFL problems are in a theoretical stage, but only experiments on quadratics are provided which makes it very unconvincing. It would be much better if the authors can provide some sort of deep learning experiments where the proper tuning of step sizes are discussed, so that we know that the benefits are real.

(v) The set of assumptions seem to be very restrictive overall, the joint $F$ needs to be quasi-strongly monotone and star-cocoercive. In practice I do not see how they can be satisfied. The synthetic experiments are constructed in a way that those conditions are met, but as I said, it is important to figure out whether the benefits are true in general, otherwise the gain would be quite restrictive.

**Questions:**

(i) What is the meaning of the solution in MpFL? I understand that it is a Nash equilibrium, but does it have any special meaning in FL systems? It seems to me a bit strange that clients in FL systems pursue conflicting goals.

(ii) How do we tune the step size of the proximal operator, and how do we select the subroutine in practice?

(iii) What is the main technical challenge in the analysis, the extension from gradient framework to proximal framework seems to be pretty straightforward, what makes it difficult/novel in this case?

---

> ### Author Response · Authors · 2025-11-24
> **Rebuttal (1/2)**
>
> We thank the reviewer for the constructive feedback and for their time and effort spent reviewing. Below, we provide responses to each point mentioned by the reviewer.
>
> ## Weaknesses
>
> > (i) The claimed better communication complexity of the PEARL-Prox algorithm seems to rely on selecting $\lambda$ optimally. However, the selection depends on $L\_\mathrm{max}$ and the condition number $\kappa$, which is hard or impossible to compute ... we need to select the local step size. In addition, the comparison happens when $\tau$ goes to infinity, which is not practical …
>
> First, regarding the reviewer’s comment on the step size selection, please note that **this is not a limitation specific to our work.**
> Most existing analyses of SGD or stochastic algorithms for minimax or multiplayer games use step sizes that depend on the condition number of the problem, total iteration number, or even the variance bound for stochastic gradients. In our view, the value of these results is that they demonstrate what is theoretically achievable, asymptotic order of convergence under a given set of assumptions, not that they propose the step size that is computable and can be used directly in practice. For generic applications of PEARL-Prox, $\lambda$ and step size will be used as hyperparameters and will be tuned, but this type of gap is prevalent in the literature (adaptive algorithms for stochastic problems often have two or more hyperparameters to tune) and arguably inevitable.
>
> Second, regarding the choice of $\tau$, note that our Corollary 4.8 more generally shows fast (linear) convergence to a neighborhood of equilibrium whose size becomes smaller as $\tau$ increases, so **even without stating that $\tau \to \infty$, we do achieve higher accuracy with larger $\tau$ (see our Figure 3).**
>
> Please note that $\tau \to \infty$ was our mathematical oversimplification for: (i) we are primarily considering the scenario where local compute budget is sufficient and sparse communication is preferred, and (ii) local proximal operations (which correspond to solving strongly convex minimization problems) can be solved with good accuracy with moderately large $\tau$. Therefore, we disagree with the reviewer’s comment that this represents the impracticality of our results. However, we thank the reviewer for pointing us to what may require further explanation, and we updated our discussion of communication efficiency on page 8 accordingly.
>
> > (ii) The proposed algorithm is novel only in the sense that it is applied in the MpFL setting … it is not unexpected that switching from PEARL-SGD to Prox yields such benefits, given the previous examples such as FedProx, ProxSkip and so on.
>
> Here we point out that despite conceptual similarity, **PEARL-Prox is not a simple re-packaging of existing results such as FedProx or ProxSkip, but is a fundamentally different type of analysis.** Furthermore, although the idea of utilizing proximal operations to mitigate drift is not new by itself, carrying out the actual technical analysis in the distinct MpFL setting is clearly a novel and important contribution. Please refer to our common response for the details.
>
> > (iii) I do not get the motivation why we are considering MpFL setting, it seems to be very related to the Personalized Federated Learning framework where proximal algorithms are known to work very well. Is there a real world problem that the setting applies to while simpler frameworks can not? It seems to me that only unnecessary mathematical complications are added to the framework, but the practical problems can already be represented by a simpler framework.
>
> We kindly ask the reviewer to view **MpFL as a general optimization framework for games where communication between players is an actual computational consideration.**
> There are numerous applications of multiplayer games in the literature, but they mostly (if not always) implicitly assume that all players have access to each other’s objectives and actions at any time, which overlooks that this may not be the case when each player is, e.g., a private organization or entity. Possible scenarios of this will include federated RL in the multi-agent setting (with applications to training game-playing AIs, autonomous driving, or robotics), multi-agent debate among large language models for improved reasoning, or financial markets with autonomous trading agents.
>
> With this perspective, MpFL is clearly different from, and more general than, personalized federated learning (PFL). It is not a purposeless mathematical complication as the reviewer believes.

---

> ### Author Response · Authors · 2025-11-24
> **Rebuttal (2/2)**
>
> > (iv) The empirical evidence seems to be very limited ... It would be much better if the authors can provide some sort of deep learning experiments ...
>
> We agree with the reviewer that an evaluation on deep learning would add practical value to the paper. However, we also believe that **not all papers must necessarily focus on large models or complex experimental settings, and this should not be considered a major weakness of our work**, where the claimed contributions are primarily theoretical. Also, quadratic games are one of the most popular testbeds for theoretical works within the minimax optimization or game theory literature.
>
> > (v) The set of assumptions seem to be very restrictive overall, the joint $F$ needs to be quasi-strongly monotone and star-cocoercive. In practice I do not see how they can be satisfied. The synthetic experiments are constructed in a way that those conditions are met, but as I said, it is important to figure out whether the benefits are true in general, otherwise the gain would be quite restrictive.
>
> The assumptions used in our analysis are not intended to be directly applied to neural networks (practical settings), and **they do not necessarily indicate that MpFL and relevant algorithms can be used only for setups where all those assumptions are met.** A number of theoretical works on FedAvg/Local SGD in classical federated learning [1, 2, 3, 4] or Local SGDA/SEG in federated minimax optimization [5, 6, 7], for example, assumed smooth and strongly convex objectives, while these algorithms are primarily used for practical applications involving deep neural networks. The goal of such theory papers, and ours as well, is to provide a solid theoretical foundation and a starting point that the community could build upon.
>
>
> [1] Stich. Local SGD converges fast and communicates little. ICLR, 2019.
>
> [2] Khaled et al. Tighter theory for local SGD … data. AISTATS, 2020.
>
> [3] Gorbunov et al. Local SGD: Unified theory and new efficient methods. AISTATS, 2021.
>
> [4] Mishchenko et al. ProxSkip: … Finally! ICML, 2022.
>
> [5] Deng & Mahdavi. Local Stochastic Gradient Descent Ascent … AISTATS, 2021.
>
> [6] Beznosikov et al. Decentralized Local Stochastic Extra-Gradient for Variational Inequalities. NeurIPS, 2022.
>
> [7] Zhang et al. Communication-Efficient … Unified Analysis and Local Updates. ICLR, 2024.
>
> ## Questions
>
> > (i) What is the meaning of the solution in MpFL? I understand that it is a Nash equilibrium, but does it have any special meaning in FL systems? It seems to me a bit strange that clients in FL systems pursue conflicting goals.
>
> Please refer to our response to item (iii) in the Weaknesses section.
>
> > (ii) How do we tune the step size of the proximal operator, and how do we select the subroutine in practice?
>
> Players can select their subroutine as *any algorithm*—e.g., SGD (with momentum), Adam variants, recent LLM optimizers such as Muon, etc.—that solves their regularized local minimization problem (for computing the proximal operator) efficiently, and the one that works best will depend on the formulation and properties of their objective functions. Step size will be associated with the subroutine and not the proximal operator itself, and its tuning can be done via a standard procedure for selecting the learning rate for optimizers.
>
> > (iii) What is the main technical challenge in the analysis, the extension from gradient framework to proximal framework seems to be pretty straightforward, what makes it difficult/novel in this case?
>
> The main idea of the analysis is to view the per-player proximal steps as the gradient play algorithm’s update direction with error due to the local updates, and to carefully control this error under stochasticity (Assumption 4.5 and Lemma 4.6). The way this local error is handled is similar to the case of prior PEARL-SGD—which is different from classical FL algorithms—but as a result of identifying and polishing a nice way to express general analysis using inexact proximal computations, our exposition is arguably cleaner than the analysis of PEARL-SGD. The way we utilize and analyze proximal operations in the context of games is different from the well-known proximal point methods (as players do not continuously adapt to others’ actions in MpFL) and, therefore, is novel and is not a direct adaptation of what exists in the literature.
> Regarding the difficulty, we are not sure what defines it or whether it is the right quantifier for the result’s significance; we rather believe that proofs that deliver desired results with clear intuition and are easy to follow are preferred.
>
> **We hope that our response has resolved the reviewer’s concerns. We believe that all points raised can be clarified as above, and are not limitations that justify recommending rejection of our work. If we have successfully addressed your concerns, please consider raising your mark. If you believe this is not the case, please let us know so that we have a chance to respond.**

---

> > ### Comment · Reviewer_SqrS · 2025-11-27
> >
> > I thank the authors for their detailed responses, it indeed solve some of my concerns.
> >
> > Specifically, I see that there are common limitations to the type of analysis, and the fact that $\tau$ being large can partially satisfies the claim. However, the better complexity is still based on the estimation of many parameters which is not easy. The justification is meaningful in the theoretical sense, but does not contradicts to my comment that the method being impractical, since the benefits is only there when certain accuracy are attained. This, in practice, cannot be easily checked which is the reason why I recommend experiments outside of the fully controlled setting. I also find the statement that when $\tau$ is large enough, it is still approximately linear converge a bit vague, since this implicitly assumes certain conditions on $\tau$ and the converge at the same time, and this will vary if we change the local solver.
> >
> > By the step size of the proximal operator I mean $\lambda$ associated with the quadratic regularizer, instead of the step size of local solvers.

---

> > > ### Author Response · Authors · 2025-12-04
> > >
> > > We thank the reviewer for engaging in further discussion.
> > >
> > > However, computing the proximal operator amounts to minimizing a strongly convex problem, and it converges quickly even with a simple SGD subroutine. As stated in Line 386 of page 8, the upper bound on the number of SGD iterations required to reach the accuracy needed for Corollary 4.8 is only 25. We believe this directly addresses the reviewer’s concern about impracticality.
> > >
> > > Regarding our claim about sufficiently large $\tau$, our meaning is precise. In the bound of Corollary 4.8, $\mathbb{E} \left[ \big\\|\mathbf{x}^R - \mathbf{x}^\star\big\\|^2 \right]
> > >     \le \left( 1 - \frac{\mu\zeta}{\lambda} \right)^R \\|\mathbf{x}^0 - \mathbf{x}^\star\\|^2 + \left( 2 + \frac{2\lambda}{\mu} \right) \frac{2\sigma^2 \log\tau}{\mu\zeta\lambda\tau}$. To have $\mathbb{E} \left[ \big\\|\mathbf{x}^R - \mathbf{x}^\star\big\\|^2 \right] \le \epsilon$, we can choose $\tau \ge \tilde{\Omega}(\epsilon^{-1})$ and $R \ge \Omega(\log \epsilon^{-1})$ to make each term in the bound at most $\frac{\epsilon}{2}$.
> > > Thus, $\tau$ can be selected as a concrete finite quantity depending on the desired accuracy $\epsilon$. We have updated lines 388–396 on page 8 to make this fully explicit, and we believe the revised explanation removes any remaining ambiguity. Although the precise lower bound on $\tau$ may vary with the local solver, we do not see why this should be viewed as a weakness of our results.
> > >
> > > As mentioned in our initial rebuttal, $\lambda$ can be treated as a hyperparameter and tuned using, for example, grid search—just as is common for many optimization methods. As a potentially more efficient alternative, one may initialize $\lambda$ at some warm-started value $\lambda_0$ and, at each communication step, multiply it by a fixed factor $\beta > 1$ whenever the norm of the joint gradient operator increases (similar in spirit to line-search–style step-size selection). This heuristic works in the settings of our experiments, and we expect it to be adequate for most practical applications.

---

### Official Review · Reviewer_iBQs · 2025-10-31

**Soundness:** 3
**Presentation:** 3
**Contribution:** 2
**Rating:** 4
**Confidence:** 3

**Summary:**

The paper tackles “player drift” in Multiplayer Federated Learning (MpFL). It introduces PEARL-Prox, which has each player solve a locally regularized objective between communication rounds. The authors provide convergence guarantees, and verify the theoretical results via numerical experiments, demonstrating the effectiveness and practicality of the proposed PEARL-Prox in handling player drift.

**Strengths:**

This paper introduces PEARL-Prox, a proximal-type algorithm designed for Multiplayer Federated Learning (MpFL), where multiple clients (players) optimize their own utility functions instead of sharing a single global objective.

PEARL-Prox lets each player optimize a regularized objective with high accuracy, ensuring convergence to the equilibrium while enabling the players to exploit their local compute budgets.

The authors provide convergence guarantees, and verify the theoretical results via numerical experiments, demonstrating the effectiveness and practicality of the proposed PEARL-Prox in handling player drift.

**Weaknesses:**

1. The proposed PEARL-Prox algorithm strongly resembles the classical FedProx method. The only substantial change is that PEARL-Prox is formulated under a multi-player objective, where each client minimizes its own loss. However, the paper does not clearly articulate how this modification leads to fundamentally new algorithmic behavior, rather than a straightforward reinterpretation of FedProx under a Nash-equilibrium setting.
2. Limited experimental scope: No real-world MpFL scenario is tested, and only a single scalar metric is plotted. No tables of quantitative results or ablations appear; thus the empirical evidence rests entirely on figures.
3. Apart from PEARL-SGD, no comparison with classical FL drift-mitigation methods adapted to MpFL is shown.
4. Strong assumptions: The paper does not discuss how restrictive these are or whether PEARL-Prox could still behave well when they fail (e.g., Assumption 4.2.) .
5. Theorem 4.4 requires $\lambda>\tfrac12(\ell+2L_{\max}\sqrt\kappa)$, but guidance on estimating $\ell$ or $\kappa$ is missing. Besides, while the paper repeatedly emphasizes the importance of λ, its practical selection strategy is never clearly specified or theoretically justified.

**Questions:**

1. How does PEARL-Prox differ fundamentally from FedProx beyond the change to a multi-player objective formulation?
2. Why does the experimental evaluation omit real-world multi-player federated learning (MpFL) settings?
3. Could the authors provide quantitative results or ablation studies, beyond the single plotted metric, to substantiate the empirical claims?
4. Why are classical FL drift-mitigation algorithms (e.g., FedProx, FedDyn, SCAFFOLD) not adapted and compared under the MpFL setup, aside from PEARL-SGD?
5. How restrictive are the theoretical assumptions (e.g., Assumption 4.2)? Does PEARL-Prox remain stable or effective when these assumptions are partially violated in practice?
6. How should λ be selected in implementation, and is there theoretical or empirical evidence supporting this choice?

---

> ### Author Response · Authors · 2025-11-24
> **Rebuttal (1/2)**
>
> We thank the reviewer for the constructive feedback and for their time and effort spent reviewing. Below, we provide responses to each point mentioned by the reviewer.
>
>
> ## Weaknesses
>
>
> > 1. The proposed PEARL-Prox algorithm strongly resembles the classical FedProx method. The only substantial change is that PEARL-Prox is formulated under a multi-player objective, where each client minimizes its own loss. However, the paper does not clearly articulate how this modification leads to fundamentally new algorithmic behavior, rather than a straightforward reinterpretation of FedProx under a Nash-equilibrium setting.
>
>
> The “only change” to the formulation of FL as a multiplayer game mentioned by the reviewer in fact **requires major technical changes in convergence proofs compared to FedProx. It is not a simple re-packaging or reinterpretation of existing results, but is a fundamentally different type of analysis.** Please refer to our common response for the details.
>
>
> > 2. Limited experimental scope: No real-world MpFL scenario is tested, and only a single scalar metric is plotted. No tables of quantitative results or ablations appear; thus the empirical evidence rests entirely on figures.
>
>
> > 3. Apart from PEARL-SGD, no comparison with classical FL drift-mitigation methods adapted to MpFL is shown.
>
> The reviewer mentions that we use a single scalar metric with no ablation. However, besides directly measuring the distance to the equilibrium $\mathbf{x}^\star$, it is unclear how the performance should be measured in the context of games. Plotting individual loss (objective) functions of each player, for example, is not a meaningful illustration of convergence.
>
> Furthermore, MpFL is a recently introduced framework whose development is primarily theoretical, and currently, there is no large-scale benchmark to run ablation on. Benchmarks and algorithms for classical federated learning, including the algorithms like FedProx or ProxSkip introduced for drift mitigation, do not take into account any meaningful game theoretic structure, and therefore, PEARL-Prox cannot be directly compared against them. In Section B of [1], it is explained in detail why comparing MpFL algorithms to classical FL algorithms is an ill-posed task.
>
> We agree that an extensive evaluation on a real application would add empirical significance to the paper. However, we also believe that **not all papers must necessarily focus on large models or complex experimental settings, and this should not be considered a major weakness of our work**, where the claimed contributions are primarily theoretical. Our experiments are mainly for the purpose of verifying the predictions from our theory.
>
> [1] Yoon et al. Multiplayer Federated Learning: Reaching Equilibrium with Less Communication. NeurIPS, 2025.
>
> > 4. Strong assumptions: The paper does not discuss how restrictive these are or whether PEARL-Prox could still behave well when they fail (e.g., Assumption 4.2.).
>
>
> The assumptions used in our analysis parallel the way a number of prior theoretical works on FedAvg/Local SGD in classical federated learning [2, 3, 4, 5] or Local SGDA/SEG in federated minimax optimization [6, 7, 8] were developed. These works assume smooth and strongly convex objectives (analogous to our Assumption 4.2), while the algorithms analyzed there are primarily used for practical applications involving deep neural networks, where these assumptions fail to hold. The goal of such theory papers, and ours as well, is to provide a solid theoretical foundation and a starting point that the community could build upon, using the analytical framework where a meaningful message can be delivered.
>
>
> [2] Stich. Local SGD converges fast and communicates little. ICLR, 2019.
>
> [3] Khaled et al. Tighter theory for local SGD … data. AISTATS, 2020.
>
> [4] Gorbunov et al. Local SGD: Unified theory and new efficient methods. AISTATS, 2021.
>
> [5] Mishchenko et al. ProxSkip: … Finally! ICML, 2022.
>
> [6] Deng & Mahdavi. Local Stochastic Gradient Descent Ascent … AISTATS, 2021.
>
> [7] Beznosikov et al. Decentralized Local Stochastic Extra-Gradient for Variational Inequalities. NeurIPS, 2022.
>
> [8] Zhang et al. Communication-Efficient … Unified Analysis and Local Updates. ICLR, 2024.

---

> ### Author Response · Authors · 2025-11-24
> **Rebuttal (2/2)**
>
> > 5. Theorem 4.4 requires $\lambda > \frac{1}{2}(\ell + 2L\_\mathrm{max} \sqrt{\kappa})$, but guidance on estimating  $\ell$ or $\kappa$ is missing. Besides, while the paper repeatedly emphasizes the importance of $\lambda$, its practical selection strategy is never clearly specified or theoretically justified.
>
>
> Let us highlight here that **the lack of strategies to estimate $\ell$ or $\kappa$ should not be seen as a limitation specific to our work.** There is a consensus in the literature that quantities like strong convexity/monotonicity parameter, which are used in the convergence analyses in a non-local manner (involving the optimum $\mathbf{x}^\star$), are very challenging to estimate [9, 10]. In our case, both parameters $\mu, \ell$ are non-local by their nature (as they involve $\mathbf{x}^\star$ in their definitions), making their estimation challenging for the same reason. To the best of our knowledge, no prior work that introduced/employed the quasi-strong monotonicity or star-cocoercivity assumptions has managed to do that.
>
> Nevertheless, we agree with the reviewer that including a discussion on how $\lambda$ should be chosen in practice will be beneficial. We view that in practice, the tuning of $\lambda$ for PEARL-Prox can be done in a way similar to, e.g., the tuning of momentum hyperparameters for optimizers like Adam. Alternatively, one could use the heuristic of initializing $\lambda$ to some warm-started $\lambda_0$ and at the next communication step, multiplying a fixed factor $\beta > 1$ to it if the norm of the joint gradient operator has increased (similar to line search approach for step size selection). This works for the settings of our experiments, and we anticipate that this will often be effective enough for practical applications.  We thank the reviewer for the suggestion, and we will add the discussion in the revision.
>
>
> [9] O'Donoghue & Candes. Adaptive Restart for Accelerated Gradient Schemes. Foundations of Computational Mathematics, 2015.
>
> [10] Su et al. A Differential Equation for Modeling Nesterov's Accelerated Gradient Method: Theory and Insights. JMLR, 2016.
>
>
> ## Questions
>
>
> > 1. How does PEARL-Prox differ fundamentally from FedProx beyond the change to a multi-player objective formulation?
>
>
> Please refer to our response to item 1 in the Weaknesses section.
>
>
> > 2. Why does the experimental evaluation omit real-world multi-player federated learning (MpFL) settings?
>
>
> Please refer to our response to items 2 & 3 in the Weaknesses section.
>
> > 3. Could the authors provide quantitative results or ablation studies, beyond the single plotted metric, to substantiate the empirical claims?
>
> Please refer to our response to items 2 & 3 in the Weaknesses section.
>
>
> > 4. Why are classical FL drift-mitigation algorithms (e.g., FedProx, FedDyn, SCAFFOLD) not adapted and compared under the MpFL setup, aside from PEARL-SGD?
>
> Please refer to our response to items 2 & 3 in the Weaknesses section.
> To repeat for emphasis: directly comparing MpFL algorithms to classical FL algorithms is an ill-posed task due to the fundamental difference in problem setups and inability to perform averaging steps—which all classical FL algorithms use—in MpFL, due to varying dimensionality of the players’ actions. Also, please refer to Section B of [1] for further details on this point.
>
>
> > 5. How restrictive are the theoretical assumptions (e.g., Assumption 4.2)? Does PEARL-Prox remain stable or effective when these assumptions are partially violated in practice?
>
>
> Assumption 4.2 is not meant to be met in practice. We do observe that PEARL-Prox converges without major stability issues in some preliminary deep learning settings constructed using the FEMNIST dataset, although did not include them in our work as we focus on the foundational theory.
>
>
> > 6. How should λ be selected in implementation, and is there theoretical or empirical evidence supporting this choice?
>
>
> Please refer to our response to item 5 in the Weaknesses section.
>
> **We hope that our response has resolved the reviewer’s concerns. We believe that all points raised can be clarified as above, and are not limitations that justify recommending rejection of our work. If we have successfully addressed your concerns, please consider raising your mark. If you believe this is not the case, please let us know so that we have a chance to respond.**

---

### Official Review · Reviewer_n5AP · 2025-11-03

**Soundness:** 2
**Presentation:** 2
**Contribution:** 2
**Rating:** 4
**Confidence:** 4

**Summary:**

The paper consider the framework of multiplayer federated learning (MpFL), where the goal is to find a Nash equilibrium of the game. A communication-efficient algorithm, PEARL-Prox, is proposed to mitigate the heterogeneity effect introduced by different client data in order to speedup the convergence, compared to PEARL-SGD. Convergence analysis and numerical simulations are provided.

**Strengths:**

1. The motivation, mitigating divergence due to excessive local computation under competitive objectives, is clear.

2. The convergence analyses are detailed. Both linear convergence for exact proximal computation and convergence to a bounded neighborhood for inexact computation are provided.

**Weaknesses:**

1. The idea of PEARL-Prox has limited novelty. While authors stress the difference from FedProx, the idea of adding a proximal term to mitigate divergence is conceptually similar. Novelty lies mostly in the game-theoretic reinterpretation rather than algorithmic mechanics.

2. The convergence analyses are straightforward. The techniques and ideas of analyses are similar to gradient play in game-theory literature, which undermines its technical difficulty.

3. The claim that PERL-Prox improves convergence rate to a linear rate is questionable. I suspect the claimed linear convergence is practically impossible as the sub minimization problem can never be solved exactly. In order words, if allowing inexactness, the convergence rate still should be $O(\epsilon^{-1/2})$, which is caused by the variance of sampling process.

4. All experiments are synthetic quadratic games and no real-world federated or multi-agent datasets are tested. Lacks evaluation of scalability (large n or high-dimensional d), robustness to noise, or comparison to strong FL baselines like FedProx/FedAvg under realistic conditions.

**Questions:**

1. Compared to FedProx, what is the novelty of the proposed algorithm (conceptually and technically)?

2. What makes the convergence analysis different from gradient play?

3. What is the convergence rate of PEARL-Prox with inexactness? Is it still linear if diminishing stepsizes are applied? Please show the rate in the order of $\epsilon$.

4. How does PEARL-Prox perform if more practical conditions are considered, e.g., non-convexity, asynchrony, partial participation, etc?

---

> ### Author Response · Authors · 2025-11-24
> **Rebuttal (1/2)**
>
> We thank the reviewer for the constructive feedback and for their time and effort spent reviewing. Below, we provide responses to each point mentioned by the reviewer.
>
> ## Weaknesses
>
> > 1. The idea of PEARL-Prox has limited novelty. While authors stress the difference from FedProx, the idea of adding a proximal term to mitigate divergence is conceptually similar. Novelty lies mostly in the game-theoretic reinterpretation rather than algorithmic mechanics.
>
> The “game-theoretic reinterpretation” mentioned by the reviewer in fact **requires major technical changes in convergence proofs compared to FedProx. It is not a simple re-packaging of existing results, but is a fundamentally different type of analysis.** Please refer to our common response for the details.
>
> > 2. The convergence analyses are straightforward. The techniques and ideas of analyses are similar to gradient play in game-theory literature, which undermines its technical difficulty.
>
> The main idea of the analysis is to view the per-player proximal steps as a gradient play direction with error due to the local updates, and to carefully control this error. At a high level, yes, the idea is simple. However, in our view, being technically complicated and difficult to understand does not define significance; we believe that proofs that deliver desired results with clear intuition and are easy to follow are rather desirable. At a technical level, how we handle the local errors, especially under stochasticity (Assumption 4.5 and Lemma 4.6) is the result of identifying and polishing toward the most concise exposition under the correct context. To the best of our knowledge, the results are novel and are not direct adaptations of what exists in the literature.
>
> > 3. The claim that PERL-Prox improves convergence rate to a linear rate is questionable. I suspect the claimed linear convergence is practically impossible as the sub minimization problem can never be solved exactly. In order words, if allowing inexactness, the convergence rate still should be $O(\epsilon^{-1/2})$, which is caused by the variance of sampling process.
>
> Here we are discussing the number of communications $R$ required for achieving $\\|\mathbf{x}^R - \mathbf{x}^\star\\|^2 = \mathcal{O}(\epsilon)$, not the number of total gradient evaluations (per-player), which will be $\tau R$ where $\tau$ is the number of local SGD iterations per communication round. The **communication acceleration to $\mathcal{O}(\log \epsilon^{-1})$ can be achieved even in the stochastic case,** provided that each player is allowed to run a large number of SGD iterations $\tau$ for solving the strongly convex minimization subroutines for proximal operator computation up to sufficiently high accuracy. Corollary 4.8 and Appendix C explain this in full detail.
>
> > 4. All experiments are synthetic quadratic games and no real-world federated or multi-agent datasets are tested. Lacks evaluation of scalability (large n or high-dimensional d), robustness to noise, or comparison to strong FL baselines like FedProx/FedAvg under realistic conditions.
>
> While we agree with the reviewer that further evaluation on real applications would be valuable, the benchmarks for classical federated learning (FedProx/FedAvg) and the corresponding problems are not formulated with a meaningful game theoretic structure, and hence are not suitable for the purpose of testing MpFL. The multi-agent tasks, such as MARL, could be more relevant, and there are several applications that we foresee. but implementation and evaluation of MpFL on those practical setups would require significant effort of a completely different nature, and are currently under exploration.
>
> On the other hand, we believe that **not all papers must necessarily focus on large models or complex experimental settings and this should not be considered a significant weakness of our work**, where the claimed contributions are primarily theoretical. Each of our experiments, as we explained in the paper, was carefully designed to highlight and verify different aspects of our theoretical results, which we view sufficient for completing the story for a theoretical work like ours.

---

> > ### Comment · Reviewer_n5AP · 2025-11-27
> >
> > Many thanks to the authors for responding to my comments. However, the rebuttal does not address my concerns.
> >
> > In particular, I am still concerned about whether $O(\log(\epsilon^{-1}))$ can be really achieved. As authors stated, this is achievable only when inner subproblem is exactly solved. Considering SGD as the solver, it means infinite number of local steps are necessary to reach $O(\log \epsilon^{-1})$ rounds, which is impossible in practice. My point is, if considering a finite $\tau < \infty$, communication rounds will become polynomial in $\epsilon^{-1}$. Moreover, in Corollary 4.8, the stepsize $\gamma = O(\log \tau / \tau)$, which approaches zero as $\tau \to \infty$. This again shows communication complexity $O(\log \epsilon^{-1})$ cannot be achieved in practice.
> >
> > For experiments, I still think the current quadratic setting is too simple and limited. At least, some practical and real-world settings should be added for the paper to be considered accepted.

---

> > > ### Author Response · Authors · 2025-11-27
> > >
> > > We thank the reviewer for engaging in discussion with us.
> > >
> > > However, the reviewer's comment that “communication complexity of $\mathcal{O}(\log \epsilon^{-1})$ cannot be achieved in practice” is not accurate.
> > > In Corollary 4.8, we have $\mathbb{E} \left[ \big\\|\mathbf{x}^R - \mathbf{x}^\star\big\\|^2 \right]
> > >     \le \left( 1 - \frac{\mu\zeta}{\lambda} \right)^R \\|\mathbf{x}^0 - \mathbf{x}^\star\\|^2 + \left( 2 + \frac{2\lambda}{\mu} \right) \frac{2\sigma^2 \log\tau}{\mu\zeta\lambda\tau}$. To have $\mathbb{E} \left[ \big\\|\mathbf{x}^R - \mathbf{x}^\star\big\\|^2 \right] \le \epsilon$, it is sufficient, e.g., to make each term in the bound at most $\frac{\epsilon}{2}$. This can be achieved if $\frac{\tau}{\log \tau} \ge \frac{1}{\epsilon} \left( 2 + \frac{2\lambda}{\mu} \right) \frac{4\sigma^2}{\mu\zeta\lambda}$, i.e., if $\tau \ge \tilde{\Omega}(\epsilon^{-1})$, where $\tilde{\Omega}$ suppresses the logarithmic factor.
> > > Thus $\tau$ can be chosen as a concrete finite number depending on $\epsilon$; it is not really taken to infinity. With such a choice of $\tau$, one can then take $R \ge \Omega(\log \epsilon^{-1})$ so that the first term of the bound is also less than $\frac{\epsilon}{2}$. We hope this clarifies the reviewer's concern. We also updated the paper to make this point more explicit and removed the limit argument involving $\tau \to \infty$.
> > >
> > > Regarding experiments, having a strong empirical foundation is clearly always a plus, but we respectfully disagree with the reviewer's comment that “some practical and real-world settings should be added for the paper to be considered accepted.” As we mentioned in the initial rebuttal, our work focuses on developing theory, and we do not believe that all papers **must** necessarily include large models or complex empirical settings. In a theory-oriented work such as ours, the role of experiments is to demonstrate that the theoretical predictions manifest as expected, and shifting the focus toward heavy empirical evaluation would not meaningfully assess the theory, and may even obscure the fundamental point the paper aims to highlight.

---

> ### Author Response · Authors · 2025-11-24
> **Rebuttal (2/2)**
>
> ## Questions
>
> > 1. Compared to FedProx, what is the novelty of the proposed algorithm (conceptually and technically)?
>
> For the technical novelty of PEARL-Prox, please refer to item 1 in the Weaknesses section.
>
> Also note that even though the idea of using proximal operators for drift mitigation is now standard in the literature, our main conceptual novelty lies in **formally quantifying the player drift phenomenon and addressing it.** While player drift in MpFL is a conceptual analogue of client drift caused by local updates, **it is different from classical client drift in the following aspects:** (i) each player $i$’s local minimizer of $f_i(\cdot; x_{-i})$ depends on others’ current actions x_{-i} unlike in classical FL, and due to this feature, (ii) the joint action $\mathbf{x}^p$ may diverge away to infinity under player drift. As the details and tendencies observed within the two drift phenomena are different, how and why proximal operations are capable of handling them also differ, which is reflected in our theoretical analysis.
>
> > 2. What makes the convergence analysis different from gradient play?
>
> The main idea of the analysis is to view the per-player proximal steps as **the gradient play direction (the negative joint operator $F(\mathbf{x}^p)$) with error due to the local updates, and to carefully control this error.** Assumption 4.5 and Lemma 4.6 are the components we use for analyzing these local errors, and the proof of Theorem 4.7 shows how they are combined altogether.
>
> > 3. What is the convergence rate of PEARL-Prox with inexactness? Is it still linear if diminishing stepsizes are applied? Please show the rate in the order of $\epsilon$.
>
> General convergence result for PEARL-Prox with inexactness (in proximal operator computation) is precisely stated in Theorem 4.7, and Corollary 4.8 provides an example of this, where players approximately compute prox operations using constant-step SGD subroutines. **The smaller the level of inexactness is (the higher the accuracy), the progression toward Nash equilibrium per communication round becomes closer to that of linear convergence.**
>
> Even when varying (diminishing) step sizes are used, provided that each player has sufficiently large local compute budget $\tau$ and the step size schedule is properly chosen, they will be able to compute the prox operations with high accuracy, and achieve up to $\mathcal{O}(\log \epsilon^{-1})$ communication complexity.
>
> > 4. How does PEARL-Prox perform if more practical conditions are considered, e.g., non-convexity, asynchrony, partial participation, etc?
>
> In the case of client dropout or asynchronous updates (with some nice assumptions on the randomness of these events), we anticipate that essentially the same analysis can be done, with the net effect of dropout being a reduction of effective progression per round by the dropout probability. For the purpose of our paper, we intend to provide a clean, foundational analysis under an idealized setup and leave the detailed extensions towards these directions open for the future.
>
>
> Please note that finding an equilibrium even in general nonconvex-nonconcave 2-player minimax games is intractable [1]. As such, some extra structural assumptions are needed to guarantee convergence to equilibrium for minimax optimization or multiplayer games in general. Our **quasi**-strong monotonicity and **star**-cocoercivity assumptions partly do that as they include classes of structured non-monotone games, but handling even more relaxed structural assumptions would be a more challenging (but interesting) future research direction that requires a distinct set of update rules and techniques, such as extragradient-type updates or multi-timescale step-sizes.
>
> [1] Diakonikolas et al. Efficient Methods for Structured Nonconvex-Nonconcave Min-Max Optimization. AISTATS, 2021.
>
>
>
> **We hope that our response has resolved the reviewer’s concerns. We believe that all points raised can be clarified as above, and are not limitations that justify recommending rejection of our work. If we have successfully addressed your concerns, please consider raising your mark. If you believe this is not the case, please let us know so that we have a chance to respond.**

---

### Official Review · Reviewer_7dGt · 2025-11-05

**Soundness:** 4
**Presentation:** 3
**Contribution:** 2
**Rating:** 4
**Confidence:** 4

**Summary:**

This paper proposes and studies a novel algorithm for the Multiplayer FL framework, an emergent framework that goes beyond the usual FL setting and aim to find personalized models that fit an equilibrium where all models are optimal given the values of all other models.
The proposed algorithm, coined PEARL-Prox, reduces the number of communication rounds required by allowing to perform more local training steps between communication rounds.
It is based on proximal updates, that are performed locally by clients.
When parameters are properly set, the local updated (even if performed approximately) allows for many local training steps while still guaranteeing fast convergence: this allows to reduce the number of communucations required in the multiplayer FL setting.
Numerical examples illustrate the theory on $n$-player games where objective functions are quadratics.

**Strengths:**

1. A new method for tackling client heterogeneity in the novel framework of multiplayer FL is proposed, which extend the range of applicability of this type of framework, extending the available algorithmic toolbox.
2. The proposed method is studied theoretically, with guarantees that show a reduction in the number of communication from $O(1/\epsilon^{-1/2})$ to $O(\log(1/\epsilon))$.
3. One strength of the method is that it can use any optimizer locally, since the theoretical results solely rely on the fact that a local optimization problem is approximately solved locally.

**Weaknesses:**

1. Although it makes sense to propose algorithm in novel frameworks, the proposed algorithm seems to be very closely related to algorithms like FedProx [1,2] or variants like 5GCS [3]. The authors made a significant effort to claim that the setting is fundamentally different from FL, but strong similarity show in assumptions 4.1-4.3 and subsequent proofs. Most results seem to be direct translations of existing results in federated learning.
2. The examples of "closely related frameworks" in FL look somewhat artificial. The interpretation in term of multi-agent reinforcement learning is vague and seems incorrect, as the authors claims this would require extending the framework to "multi-objective games".
3. At the end of page 4, it seems that authors claim that the player drift phenomenon is somewhat unexpected: it seems that this is very close to the client drift phenomenon, and that it arises in very similar situations.
4. Experiments look somewhat artificial, and only exhibit marginal gains w.r.t. algorithms that use only one local step. This does not look as spectacular as what the theoretical claims state.

[1] Federated Optimization in Heterogeneous Networks, Tian Li, Anit Kumar Sahu, Manzil Zaheer, Maziar Sanjabi, Ameet Talwalkar, Virginia Smith, MLSys, 2020.

{2] On Convergence of FedProx: Local Dissimilarity Invariant Bounds, Non-smoothness and Beyond, Xiao-Tong Yuan, Ping Li, NeurIPS 2022.

[2] Can 5th Generation Local Training Methods Support Client Sampling? Yes!, Michał Grudzien, Grigory Malinovsky, Peter Richtárik, AISTATS 2023.

**Questions:**

1. What are the actual challenges in extending results from the classical FL setting to multiplayer FL?
2. Are there other classical numerical benchmarks for multiplayer FL where the proposed method could be studied?
3. It seems that player drift has fundamental involves fundamental differences with the classical client drift: what are these fundamental differences?

---

> ### Author Response · Authors · 2025-11-24
>
> We thank the reviewer for the constructive feedback and for their time and effort spent reviewing. Below, we provide responses to each point mentioned by the reviewer.
>
> ## Weaknesses
>
> > 1. Although it makes sense to propose algorithm in novel frameworks, the proposed algorithm seems to be very closely related to algorithms like FedProx or variants like 5GCS … Most results seem to be direct translations of existing results in federated learning.
>
> Our results are **not** direct translations of these existing results. Please refer to our common response for the details. The case of 5GCS is similar—it takes the sum of dual iterates and uses it for global model update, implicitly assuming that all clients have iterates of the same dimension and collaborate toward minimizing the finite sum.
>
> > 2. The examples of "closely related frameworks" … The interpretation in terms of multi-agent reinforcement learning is vague and seems incorrect …
>
> In the paper, we explained that multi-agent RL setup is a related but different topic, mentioning that “MpFL does not immediately subsume this formulation in its basic form”. This is a potential connection that can be explored further in the future, and at the moment, we do not claim that multi-agent RL is an example of MpFL.
>
> > 3. At the end of page 4, it seems that authors claim that the player drift phenomenon is somewhat unexpected: it seems that this is very close to the client drift phenomenon, and that it arises in very similar situations.
>
> We are not sure which part should be interpreted as a claim that the player drift is unexpected. At the end of page 4, we simply state that player drift may occur even when the game dynamics are favorable (i.e., when $F$ is strongly monotone).
>
> While player drift is indeed a conceptual analogue of client drift caused by local updates, **it is different from client drift in the following aspects:** (i) each player $i$’s local minimizer of $f_i(\cdot; x_{-i})$ depends on others’ current actions $x\_{-i}$ unlike in classical FL, and due to this feature, (ii) the joint action $\mathbf{x}^p$ may diverge away to infinity under player drift. Therefore, **formally quantifying and addressing player drift is a novel and meaningful contribution independent of prior works on client drift.**
>
> > 4. Experiments look somewhat artificial, and only exhibit marginal gains ... This does not look as spectacular as what the theoretical claims state.
>
> Please note that our experiments, as we mention in the paper, primarily verifies the theory. Each of them was carefully selected to highlight different aspects of our theoretical results and complete the storyline. For instance, the experiments of Figure 4 were designed to show PEARL-Prox resolving divergence (player drift) when using a large number of local iterations $\tau$, rather than emphasizing the communication gain.
>
> Regarding the reviewer’s specific point on the benefits of using local steps, the effect of increasing $\tau$ can be observed in, e.g., Figure 3, where $\approx 10$ times increase in $\tau$ provides $\approx 10$ times better performance, as our Corollary 4.8 predicts. Note that the bound in Corollary 4.8 is improved as $\tau$ grows, and the communication complexity of $\Theta\left(\frac{1}{\epsilon}\right)$ is achieved in the limit of $\tau \to \infty$ (large local compute budget), while we only test a moderate range of $\tau$ for conceptual illustrations in our experiments.
>
> ## Questions
>
> > 1. What are the actual challenges in extending results from the classical FL setting to multiplayer FL?
>
> Please refer to item 1 in the Weakness section.
>
> > 2. Are there other classical numerical benchmarks for multiplayer FL where the proposed method could be studied?
>
> MpFL is a recently proposed concept (mainly theoretical), and there is no large-scale benchmark specifically designed for MpFL at the moment. Also, existing benchmarks for classical FL are not formulated with a meaningful game-theoretic structure and are not suitable for testing MpFL.
>
> On the other hand, MpFL can be applied to any multiplayer game setting where communication between players is expensive—in that regard, we view our work also as a part of the minimax optimization or multiplayer games literature. The experimental setup we use here is very commonly used in those relevant work.
>
> > 3. It seems that player drift has fundamental differences with the classical client drift: what are these fundamental differences?
>
> This is indeed the case, as we answered in our response to item 3 in the Weakness section. We appreciate the reviewer’s thoughtful comment.
>
> **We hope that our response has resolved the reviewer’s concerns. We believe that all points raised can be clarified as above, and are not limitations that justify recommending rejection of our work. If we have successfully addressed your concerns, please consider raising your mark. If you believe this is not the case, please let us know so that we have a chance to respond.**

---

### Author Response · Authors · 2025-11-24
**Common response**

We thank all reviewers for providing constructive and thoughtful feedback.
While the reviewers generally agree that the motivation of mitigating player drift is clear and that our theoretical results are solid, several comments raise concerns about the technical novelty of our work. We would like to clarify that these concerns stem from misunderstandings of our contributions, and we address them directly below.

## Distinction from FedProx and technical novelty

The FedProx algorithm [1] considers the finite sum minimization setting (classical FL) where the objective is simply to minimize $\frac{1}{N} \sum\_{i=1}^N f\_i(x)$. Here each player’s objective function $f\_i$ *does not depend on other players’ local models,* and in the aggregation (synchronization) step, the *server averages the local models $x_i$.* MpFL setting is completely different. Each player has a distinct action/model $x_i$ of generally different dimensions, so **averaging is not possible due to this domain mismatch, and players’ actions affect the objectives of each other.**

*Consequently, proof techniques from classical FL do not carry over to the MpFL setting, just as the analysis of SGD (for minimization) and the analysis of SGDA for min-max games are different*—e.g., for games, proofs can be written solely in terms of joint gradient operators without appealing to function value suboptimality $f(x\_k) - f\_*$ as most SGD analyses do. Additionally, when handling local updates, as we cannot use averaging (which provides benefits such as linear speedup in the number of clients for classical FL), we need a different argument to bound the errors due to local update steps. The similarity between PEARL-Prox and FedProx is only superficial—both algorithms use proximal operations. Beyond that, the MpFL setting required us to introduce novel technical components, which significantly differ from prior work.

[1] Li et al. Federated Optimization in Heterogeneous Networks. MLSys, 2020.

---

### Meta-Review · Area_Chair_j2S7 · 2025-12-22

**Summary:**

The paper proposed PEARL-Prox, a method for handling player drift in multiplayer federated learning (MpFL). The reviewers appreciated the studied problem and the paper's rigor.

However, the paper received negative scores only. In particular, all reviewers raised concerns about similarities of PEARL-Prox to classic FedProx and regarding the experimental validation.

Overall, the paper looks promising, but might benefit from a rewrite in terms of the comparisons to FedProx and adding in more realistic experimental settings.

**Reviewer Concerns:**

The authors did respond via extensive rebuttals. They argued that while their method is related to FedProx, the actual technical analysis is very different, in particular evoking techniques from gradient play. They also argued that experiments in the context of deep learning are outside the scope of their work.

Two reviewers (n5AP and SqrS) responded. Reviewer n5AP stated that their concerns, in particular regarding limited experimental evaluation, remain. Reviewer SqrS stated that some of their concerns were addressed, however remained unconvinced by the empirical evaluation.

**Reviewer Scores:**

Overall, since all reviewers argued about limited experimental evaluation and the two responses from reviewers reiterated these concerns, it is unlikely that the discussion would have led to a change in position regarding the experimental validation. Thus, while reviewer SqrS may have raised their score slightly (due to acknowledging that some of their concerns were addressed), I do not expect that the other reviewers would have raised scores significantly.

---

### Decision · Program_Chairs · 2026-01-26

Reject